# BATTLE: Towards Behavior-oriented Adversarial Attacks against Deep Reinforcement Learning

## Abstract

Evaluating the performance of deep reinforcement learning (DRL) agents under adversarial attacks that aim to induce specific behaviors, i.e., behavior-oriented adversarial attacks, is crucial for understanding the robustness of DRL agents. Prior research primarily focuses on directing agents towards pre-determined states or policies, lacking generality and flexibility. Therefore, it is important to devise universal attacks that target inducing specific behaviors in a victim. In this work, we propose BATTLE, a universal behavior-oriented adversarial attack method. In BATTLE, an intention policy is trained to align with human preferences for flexible behavior orientation, while the adversary is trained to guide the victim policy to imitate the intention policy. To improve the attack performance, we introduce a weighting function that assigns importance weights over each state. Our empirical results over several manipulation tasks of Meta-world show the superiority of BATTLE in behavior-oriented adversarial attack settings, outperforming current adversarial attack algorithms. Furthermore, we also demonstrate that BATTLE can improve the robustness of agents under strong attacks by training with adversary. Lastly, we showcase the strong behavior-inducing capability of BATTLE by guiding Decision Transformer agents to act in line with human preferences across various MuJoCo tasks. Our videos are available in https://sites.google.com/view/jj9uxjgmba5lr3g.

## 1 Introduction

Reinforcement learning (RL) (Sutton & Barto, 2018) combined with deep neural networks (DNN) (Le-Cun et al., 2015) shows extraordinary capabilities of allowing agents to master complex behaviors in various domains. However, recent findings (Huang et al., 2017; Pattanaik et al., 2018; Zhang et al., 2020) reveal that well-trained RL agents parameterized by DNN suffer from vulnerability against test-time attacks, raising concerns in high-risk or safety-critical situations. To understand adversarial attacks on learning algorithms and enhance the robustness of DRL agents, it is crucial to evaluate the performance of the agents under any potential adversarial attacks with certain constraints. In other words, identifying a universal and strong adversary is essential.

Two main challenges persist in devising effective universal and strong attacks. Firstly, existing strategies, which primarily aim at diminishing cumulative rewards, fall short in specifying explicit attack targets. Prior research (Zhang et al., 2020; 2021; Sun et al., 2022) considers training strong adversary by perturbing state observations of victim to obtain the worst case expected return. Nevertheless, it might be more practical to setup both a reward function and constraints, rather than encoding the complex safety requirements directly into rewards (Achiam et al., 2017; Gu et al., 2022; Vamplew et al., 2022), for many applications of RL. Therefore, only quantifying the decrease in cumulative reward can be too generic and result in limited attack performance

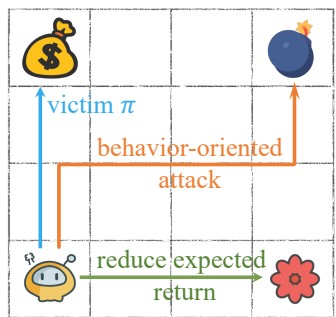

Figure 1: An example illustrating the distinction between our approach and generic attacks.

when adversaries target specific safety attacks. Consider the scenario depicted in Figure 1, where a

robot is tasked with collecting coins. Previous attack methods aim at inducing the robot away from the coins by minimizing the expected return. In contrast, it might be interested in causing specific unsafe behaviors, such as inducing the robot to collide with a bomb. Secondly, Predefined targets are usually rigid and inefficient. Another line of work (Hussenot et al., 2019a; Lin et al., 2017b) primarily focuses on misleading the agent towards a predetermined goal state or target policy, overlooking specific behaviors. Additionally, the difficulty and expense of providing a well-designed targeted policy result in these methods lacking generality and flexibility. In a broader sense, these adversarial attacks are incapable of controlling the behaviors of agents as a form of universal attack.

To tackle these challenges, we present a novel adversarial attack method, namely BATTLE, which focuses on **B**ehavior-oriented **A**dversarial a**TT**acks against deep r**E**inforcement learning agents. At its core, BATTLE employs an adversary to perturb the victim agent's observations while leveraging an intention policy for step-by-step guidance for the victim to imitate. Instead of relying on a predefined target policy, the intention policy is trained to align with human intent as a flexible behavior orientation during adversary training. Furthermore, we incorporate a weighting function to capture remarkable moments through state re-weighting, improving overall performance and efficiency. Benefiting from recent progress in preference-based reinforcement learning (PbRL) (Lee et al., 2021a; Park et al., 2022; Liang et al., 2022; Liu et al., 2022), our method facilitates the behavior of victim agent to be aligned with human intentions. Consequently, BATTLE's adversary effectively leads the victim into executing human-desired behaviors through iterative refinement.

In summary, our contributions fall into four categories. Firstly, we propose a universal behavior-oriented adversarial attack method against DRL agents, designed to effectively induce specific behaviors in a victim agent. Secondly, we theoretically analyze BATTLE and provide a convergence guarantee under only mild conditions. Thirdly, we test in multiple scenarios and experiments on Meta-world that demonstrate BATTLE outperforms the baselines by a large margin. Empirical results demonstrate that both online and offline RL agents are vulnerable to our proposed adversarial attacks, including the most recent Decision Transformer. Finally, we considerably enhance the robustness of DRL agents by learning with BATTLE attacker in adversarial training.

## 2 RELATED WORK

Previous works on adversarial attacks study the vulnerability of a DRL agent. Huang et al. (2017) computes adversarial perturbations via utilizing the technique of FGSM (Goodfellow et al., 2015) to mislead the victim policy, not to choose the optimal action. Pattanaik et al. (2018) presents an approach that leads the victim to select the worst action based on the Q-function of the victim. Gleave et al. (2020) conducts adversarial attacks under the two-player Markov game instead of perturbing the agent's observation. Zhang et al. (2020) proposes the state-adversarial MDP (SA-MDP) and develops two adversarial attack methods named Robust Sarsa (RS) and Maximal Action Difference (MAD). SA-RL (Zhang et al., 2021) directly optimizes the adversary to perturb state in the form of end-to-end RL. PA-AD (Sun et al., 2022) designs an RL-based "director" to find the optimal policy perturbing direction and construct an optimized-based "actor" to craft perturbed states according to the given direction. While untargeted adversarial attacks aim to cause the victim policy to fail, our method emphasizes manipulating the behaviors of the victim. In other words, the perturbed behaviors of the victim align with the manipulator's preferences. Another line of works (Pinto et al., 2017; Mandlekar et al., 2017; Pattanaik et al., 2018) consider using adversarial examples to improve the robustness of policies, although it is out of the scope of this paper.

There are a few prior works that focus on inducing DRL agents to pre-determined states or policies. Lin et al. (2017a) first proposes a targeted adversarial attack method against DRL agents, which attacks the agent to reach a targeted state. Buddareddygari et al. (2022) also present a strategy to mislead the agent towards to a specific state by placing an object in the environment. The hijacking attack (Boloor et al., 2020) is proposed to attack agents to perform targeted actions on autonomous driving systems. Hussenot et al. (2019b) provides a new perspective that attacks the agent to imitate a target policy. Lee et al. (2021b) investigates targeted adversarial attacks against the action space of the agent. Our method differs that we train an intention policy to serve as flexible behavior orientation, rather than relying on a predetermined target state or policy. Consequently, BATTLE can effectively lead the victim policy to perform human desired behaviors.

Training agents with human feedback has been investigated in several works. PbRL provides an effective way to utilize human preferences for agent learning. Christiano et al. (2017) proposes a basic

learning framework for PbRL. To further improve feedback efficiency, Ibarz et al. (2018) additionally utilizes expert demonstrations to initialize the policy besides learning the reward model from human preferences. However, previous methods need plenty of human feedback, which is usually impractical. Many recent works have proposed to tackle this problem. Lee et al. (2021a) presents a feedback-efficient PbRL algorithm, which benefits from unsupervised exploration and reward relabeling. Park et al. (2022) further improves feedback efficiency by semi-supervised reward learning and data augmentation, while Liang et al. (2022) proposes an intrinsic reward to enhance exploration. Liu et al. (2022) continues to improve the feedback efficiency by aligning the Q-function with human preferences. To the best of our knowledge, our method is the first to conduct a behavior-oriented adversarial attack against DRL agents through PbRL.

## 3 PROBLEM SETUP

**The Victim Policy**. In RL, agent learning can be modeled as a finite horizon Markov Decision Process (MDP) defined as a tuple $(\mathcal{S}, \mathcal{A}, \mathcal{R}, \mathcal{P}, \gamma)$. $\mathcal{S}$ and $\mathcal{A}$ denote state and action space, respectively. $\mathcal{R} : \mathcal{S} \times \mathcal{A} \times \mathcal{S} \to \mathbb{R}$ is the reward function and $\gamma \in (0, 1)$ is the discount factor. $\mathcal{P} : \mathcal{S} \times \mathcal{A} \times \mathcal{S} \to [0, 1]$ denotes the transition dynamics, which determines the probability of transferring to $\mathbf{s}'$ given state $\mathbf{s}$ and action $\mathbf{a}$. We denote the stationary policy $\pi_\nu : \mathcal{S} \to \mathcal{P}(\mathcal{A})$, where $\nu$ are parameters of the victim. We suppose the victim policy is fixed and uses the approximator.

**The Adversarial Policy.** To study behavior-oriented adversarial attack with human preferences, we formulate it as rewarded state-adversarial Markov Decision Process (RSA-MDP). Formally, a RSA-MDP is a tuple $(\mathcal{S}, \mathcal{A}, \mathcal{B}, \widehat{\mathcal{R}}, \mathcal{P}, \gamma)$. The adversary $\pi_\alpha : \mathcal{S} \to \mathcal{P}(\mathcal{S})$ perturbs the states before the victim observes them, where $\alpha$ are parameters of the adversary. The adversary perturbs the state $\mathbf{s}$ into $\tilde{\mathbf{s}}$ restricted by $\mathcal{B}(\mathbf{s})$ (i.e., $\tilde{\mathbf{s}} \in \mathcal{B}(\mathbf{s})$). $\mathcal{B}(\mathbf{s})$ is defined as a small set $\{\tilde{\mathbf{s}} \in \mathcal{S} : \| \mathbf{s} - \tilde{\mathbf{s}} \|_p \leq \epsilon \}$, which limits the attack power of the adversary and $\epsilon$ is attack budget. Since directly generating $\tilde{\mathbf{s}} \in \mathcal{B}(\mathbf{s})$ is hard, the adversary learns to produce a Gaussian noise $\Delta$ with $\ell_\infty(\Delta)$ less than 1, and we obtain the perturbed state through $\tilde{\mathbf{s}} = \mathbf{s} + \Delta * \epsilon$. The victim takes action according to the observed $\tilde{\mathbf{s}}$, while true states in the environment are not changed. $\pi_{\nu \circ \alpha}$ denotes the perturbed policy, which is victim policy under the adversarial attack. Unlike SA-MDP (Zhang et al., 2020), RSA-MDP introduces $\widehat{\mathcal{R}}$, which learns from human preferences. The target of RSA-MDP is to solve the optimal adversary $\pi_\alpha^*$, which enables the victim to achieve the maximum cumulative reward (i.e., from $\widehat{\mathcal{R}}$) over all states. Lemma C.1 shows that solving the optimal adversary in RSA-MDP is equivalent to finding the optimal policy in MDP $\hat{\mathcal{M}} = (\mathcal{S}, \hat{\mathcal{A}}, \widehat{\mathcal{R}}, \widehat{\mathcal{P}}, \gamma)$, where $\hat{\mathcal{A}} = \mathcal{S}$ and $\widehat{\mathcal{P}}$ is the transition dynamics of the adversary.

## 4 METHOD

In this section, we introduce our method BATTLE. The core idea of BATTLE is twofold: firstly, it learns an intention policy that acts as the learning target for the adversarial policy, effectively guiding the victim towards behaving human-desired behavior. Secondly, we introduce a weighting function to enhance the adversary's performance and formulate BATTLE as a bi-level optimization problem. The framework of BATTLE is shown in Figure 2 and detailed procedure is summarized in Appendix A.

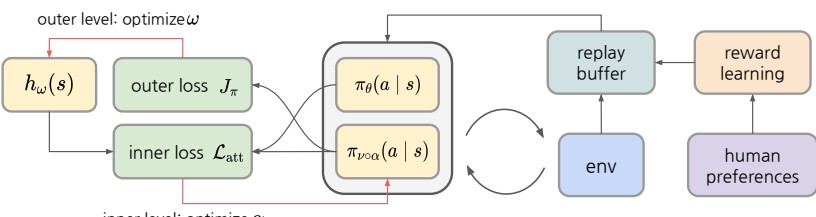

Figure 2: Overview of BATTLE. BATTLE jointly learns an intention policy $\pi_\theta$, an adversary $\pi_\alpha$ and a weighting function $h_\omega$ under bi-level optimization framework. In the inner-level, the adversary is optimized such that $\pi_{\nu \circ \alpha}$ approaches the intention policy which learns via PbRL. In the outer-level, the weighting function is updated to improve the performance of the adversary evaluated by the outer loss $J_\pi$. $\pi_{\nu \circ \alpha}$ denotes the perturbed policy, which is victim policy under the adversarial attack.

## 4.1 LEARNING INTENTION POLICY

BATTLE aims to find the optimal adversary that manipulates the victim's behaviors to be consistent with human intentions. However, the victim policy is pre-trained for a specific task, directly learning an adversary suffer from exploration problem caused by the restriction of victim policy, making it hard to find an optimal adversary efficiently. Therefore, we introduce an intention policy $\pi_\theta$ which has unrestricted exploration space to guide adversarial policy training.

To conduct targeted attack and avoid reward engineering, we align the intention policy with human intent via PbRL, which is shown in Figure 3. In PbRL, the agent have no access to the ground-truth reward function. Humans provide preference labels between two agent trajectories and the reward function $\widehat{r}_\psi$ learns to align with the preferences (Christiano et al., 2017).

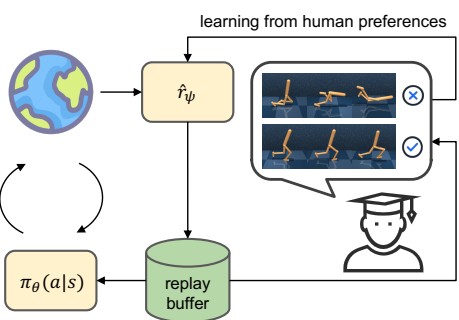

learning from human preferences

Formally, a segment $\sigma$ of length $k$ is denoted as a sequence of states and actions $\{\mathbf{s}_{t+1}, \mathbf{a}_{t+1}, \cdots, \mathbf{s}_{t+k}, \mathbf{a}_{t+k}\}$. Given a pair of segments $(\sigma^0, \sigma^1)$, human indicate which segment is preferred, where $y \in \{(0,1), (1,0), (0.5, 0.5)\}$. Following Bradley-Terry model (Bradley & Terry, 1952), a preference predictor is constructed in (1):

Figure 3: Diagram of preference-based RL.

$$P_\psi[\sigma^0 \succ \sigma^1] = \frac{\exp \sum_t \widehat{r}_\psi(\mathbf{s}_t^0, \mathbf{a}_t^0)}{\sum_{i \in \{0,1\}} \exp \sum_t \widehat{r}_\psi(\mathbf{s}_t^i, \mathbf{a}_t^i)}, \tag{1}$$

where $\sigma^0 \succ \sigma^1$ denotes $\sigma^0$ is preferred to $\sigma^1$. This predictor indicates the probability that a segment is preferred is proportional to its exponential return. Then, the reward function is optimized by aligning the predicted preference labels with human preferences through cross-entropy loss:

$$\mathcal{L}(\psi) = - \mathop{\mathbb{E}}_{(\sigma^0, \sigma^1, y) \sim \mathcal{D}} \Big[ y(0) \log P_\psi[\sigma^0 \succ \sigma^1] + y(1) \log P_\psi[\sigma^1 \succ \sigma^0] \Big], \tag{2}$$

where $\mathcal{D}$ is a dataset of triplets $(\sigma^0, \sigma^1, y)$ consisting of segment pairs and human preference labels. By minimizing (2), we obtain a reward function estimator $\widehat{r}_\psi$, which is used to provide estimated rewards for agent learning via any RL algorithms. Following PEBBLE (Lee et al., 2021a), we use an off-policy actor-critic method SAC (Haarnoja et al., 2018) to learn a well-performing policy. Specifically, the Q-function $Q_\phi$ is optimized by minimizing the Bellman residual:

$$J_Q(\phi) = \mathop{\mathbb{E}}_{\tau_t \sim \mathcal{B}} \left[ \left( Q_\phi(\mathbf{s}_t, \mathbf{a}_t) - \widehat{r}_t - \gamma \bar{V}(\mathbf{s}_{t+1}) \right)^2 \right], \tag{3}$$

where $\bar{V}(\mathbf{s}_t) = \mathbb{E}_{\mathbf{a}_t \sim \pi_\theta} \left[ Q_{\bar{\phi}}(\mathbf{s}_t, \mathbf{a}_t) - \mu \log \pi_\theta(\mathbf{a}_t | \mathbf{s}_t) \right]$, $\tau_t = (\mathbf{s}_t, \mathbf{a}_t, \widehat{r}_t, \mathbf{s}_{t+1})$ is the transition at time step $t$, $\bar{\phi}$ is the parameter of the target soft Q-function. The policy $\pi_\theta$ is updated by minimizing (4):

$$J_\pi(\theta) = \mathbb{E}_{\mathbf{s}_t \sim \mathcal{B}, \mathbf{a}_t \sim \pi_\theta} \Big[ \mu \log \pi_\theta(\mathbf{a}_t | \mathbf{s}_t) - Q_\phi(\mathbf{s}_t, \mathbf{a}_t) \Big], \tag{4}$$

where $\mu$ is the temperature parameter. By learning an intention policy, BATTLE tackles restricted exploration problem and provides an attack target for the following adversary training.

## 4.2 LEARNING ADVERSARIAL POLICY AND WEIGHTING FUNCTION

To make the victim policy perform human desired behaviors, BATTLE learns the adversary by minimizing the KL divergence between the perturbed policy $\pi_{\nu \circ \alpha}$ and the intention policy $\pi_\theta$. However, different states may have various importance to induce the victim policy to the target. To stabilize training process and improve the performance of the adversary, we introduce a weighting function $h_\omega$ to re-weight states in adversary training.

We formulate BATTLE as a bi-level optimization algorithm, which alternately updates the adversarial policy $\pi_\alpha$ and the weighting function $h_\omega$ through inner and outer optimization. In the inner level, BATTLE optimizes parameters $\alpha$ with the importance weights outputted by a weighting function $h_\omega$, and optimizes parameters $\omega$ in the outer level according to the performance of the adversary. Intuitively, the adversary is optimized such that $\pi_{\nu \circ \alpha}$ approaches the intention policy in the inner

level, while the weighting function learns to improve the adversary performance by evaluating the performance of the adversary through a meta-level loss $J_\pi$ in (7). The whole objective of BATTLE is:

$$
\begin{aligned}
\min_{\omega} \quad & J_\pi(\alpha(\omega)), \\
\text{s.t.} \quad & \alpha(\omega) = \arg\min_{\alpha} \mathcal{L}_{\text{att}}(\alpha; \omega, \theta).
\end{aligned}
\tag{5}
$$

**Inner-level Optimization: Training adversarial policy** $\pi_\alpha$. In the inner-level optimization, given the intention policy $\pi_\theta$ and the weighting function $h_\omega$, we hope to find the optimal adversarial policy by minimizing the re-weighted KL divergence between $\pi_{\nu\circ\alpha}$ and $\pi_\theta$ in (6):

$$
\mathcal{L}_{\text{att}}(\alpha; \omega, \theta) = \mathop{\mathbb{E}}_{\mathbf{s}\sim\mathcal{B}} \left[ h_\omega(\mathbf{s}) D_{\text{KL}}\left(\pi_{\nu\circ\alpha}(\mathbf{s}) \parallel \pi_\theta(\mathbf{s})\right) \right],
\tag{6}
$$

where $h_\omega(\mathbf{s})$ is the importance weights outputted by the weighting function $h_\omega$. Intuitively, the adversarial policy is optimized to make the perturbed policy be close to the intention policy, while $h_\omega$ assigns different weights to states of various importance. With the collaborative assistance of the intention policy and the weighting function, BATTLE efficiently learns an optimal adversarial policy.

**Outer-level Optimization: Training weighting function** $h_\omega$. In the outer-level optimization, we need to find a precise weighting function to balance the state distribution and assign proper weights to propel adversary learning. The weighting function is trained to distinguish the importance of states by evaluating the performance of the perturbed policy. Specifically, the perturbed policy $\pi_{\nu\circ\alpha}$ is evaluated using a policy loss in (7), which is adapted from the policy loss in (4):

$$
J_\pi(\alpha(\omega)) = \mathbb{E}_{\mathbf{s}_t\sim\mathcal{B}, \mathbf{a}_t\sim\pi_{\nu\circ\alpha(\omega)}} \left[ \mu \log \pi_{\nu\circ\alpha(\omega)}(\mathbf{a}_t|\mathbf{s}_t) - Q_\phi(\mathbf{s}_t, \mathbf{a}_t) \right],
\tag{7}
$$

where $\alpha(\omega)$ denotes $\alpha$ implicitly depends on $\omega$. Therefore, BATTLE calculates the implicit derivative of $J_\pi(\alpha(\omega))$ with respect to $\omega$ and finds the optimal $\omega^*$ by optimizing (7). To make it feasible, we make an approximation of $\arg\min_\alpha$ with the one-step gradient update. (8) obtains an estimated $\arg\min_\alpha$ with one-step updating and builds a connection between $\alpha$ and $\omega$:

$$
\hat{\alpha}(\omega) \approx \alpha_t - \eta_t \left. \nabla_\alpha \mathcal{L}_{\text{att}}(\alpha; \omega, \theta) \right|_{\alpha_t}.
\tag{8}
$$

According to the chain rule, the gradient of the outer loss with respect to $\omega$ can be expressed as:

$$
\begin{aligned}
\left. \nabla_\omega J_\pi(\alpha(\omega)) \right|_{\omega_t} &= \left. \nabla_{\hat{\alpha}} J_\pi(\hat{\alpha}(\omega)) \right|_{\hat{\alpha}_t} \left. \nabla_\omega \hat{\alpha}_t(\omega) \right|_{\omega_t} \\
&= \sum_{\mathbf{s}} f(\mathbf{s}) \cdot \left. \nabla_\omega h(\mathbf{s}) \right|_{\omega_t},
\end{aligned}
\tag{9}
$$

where $f(\mathbf{s}) = -\eta_t \cdot (\nabla_{\hat{\alpha}} J_\pi(\alpha(\omega)))^\top \nabla_\alpha D_{\text{KL}}(\pi_{\nu\circ\alpha}(\mathbf{s}) \parallel \pi_\theta(\mathbf{s}))$ and detailed derivation can be found in Appendix B. The key to obtain this meta gradient is building and computing the relationship between $\alpha$ and $\omega$. Obtaining the implicit derivative, BATTLE updates the parameters of the weighting function by taking gradient descent with outer learning rate.

In addition, we theoretically analyze the convergence of BATTLE in Theorem D.2 and D.4. In Theorem D.2, we demonstrate the convergence rate of the outer loss, i.e. the gradient of the outer loss with respect to $\omega$ will convergence to zero. Thus BATTLE learns a more powerful adversary using importance weights outputted by the optimal weighting function. In Theorem D.4, we prove the convergence of the inner loss. The inner loss of BATTLE algorithm converges to critical points under some mild conditions, which ensures the parameters of the adversary can converge to the optimal parameters. Theorems and proofs can be found in Appendix D.

## 5 EXPERIMENTS

In this section, we evaluate our method on several robotic simulated manipulation tasks from Meta-world (Yu et al., 2020) and continuous locomotion tasks from MuJoCo (Todorov et al., 2012). Specifically, our experiment contains two essential phases. In the first phase, we verify the efficacy of the proposed method through two scenarios: manipulation and opposite behaviors. Furthermore, we show the capability of our approach by fooling a popular offline RL method, Decision Transformer (Chen et al., 2021), into acting specific behaviors in the second phase. The detailed description of experiments is provided in Appendix F.

### 5.1 SETUP

**Compared Methods.** Random attack and two state-of-the-art evasion attack methods are used for comparison.

- Random: a baseline that samples random perturbed observations via a uniform distribution.

- SA-RL (Zhang et al., 2021): this method learns an adversarial policy in the form of end-to-end RL formulation.

- PA-AD (Sun et al., 2022): this method combines RL-based "director" and non-RL "actor" to find state perturbations, which is the state-of-the-art adversarial attack algorithm against DRL.

- BATTLE: our proposed method, which collaboratively learns adversarial policy and weighting function with the guidance of intention policy.

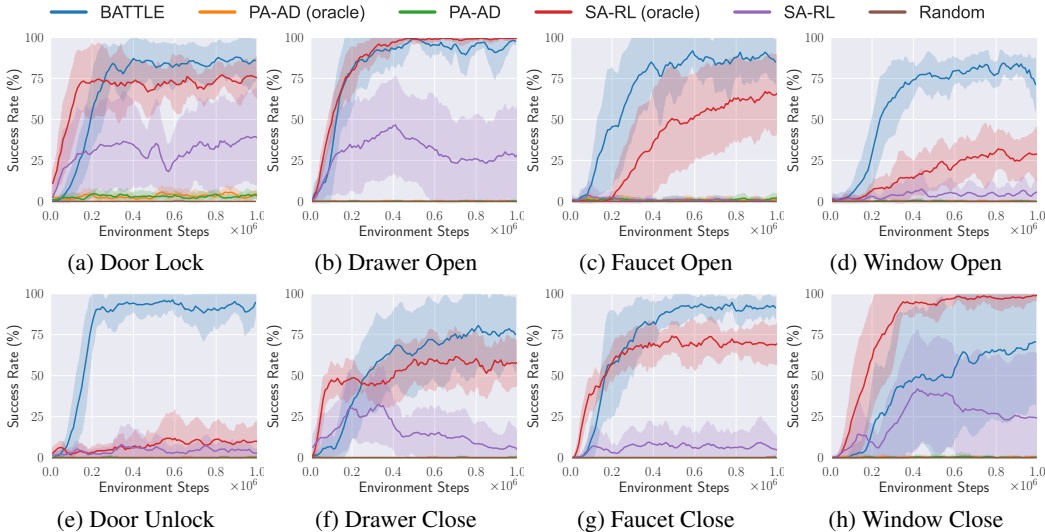

Figure 4: Training curves of different methods on various tasks in the manipulation scenario. The solid line and shaded area denote the mean and the standard deviation of success rate, respectively, over ten runs. The blue line (our method) outperforms all the baselines in PbRL setting and even exceeds most baselines in oracle setting.

**Implementation Settings.** We compare BATTLE with existing state-of-the-art adversarial attack methods. To achieve fair comparison, we make simple adjustments for SA-RL and PA-AD to suit our settings in the experiments. In their original version, both of these two methods use the negative value of the reward obtained by the victim to train an adversary. We replace it with the same estimated reward function $\widehat{r}_\psi$ as our method uses, which means they also learn from human preferences. Following the settings in PEBBLE (Lee et al., 2021a), we use a scripted teacher that provides ground truth preference labels. More details of scripted teacher and preference collection can be found in Appendix E. For the implementation of SA-RL[1] and PA-AD[2], we use the released official codebase. For fair comparison, all methods learned via PbRL are given the same number of preference labels. In the manipulation scenario, we use 9000 labels for all tasks. In the opposite behaviors scenario, we use 1000 for Window Close, 3000 for Drawer Close, 5000 for Faucet Open, Faucet Close and Window Open, 7000 for Drawer Open, Door Lock and Door Unlock. Also, to reduce the impact of PbRL, we additionally add oracle versions of SA-RL and PA-AD, which uses the ground-truth rewards of the targeted task.

We use the same experimental settings (i.e., hyper-parameters, neural networks) concerning reward learning for all methods. We quantitatively evaluate all methods by comparing the success rate of final manipulation, which is well-defined in Meta-world (Yu et al., 2020) for the opposite behaviors scenario, and we rigorously design for the manipulation scenario. As in most existing research (Zhang

---

[1]https://github.com/rll-research/BPref
[2]https://github.com/umd-huang-lab/paad_adv_rl

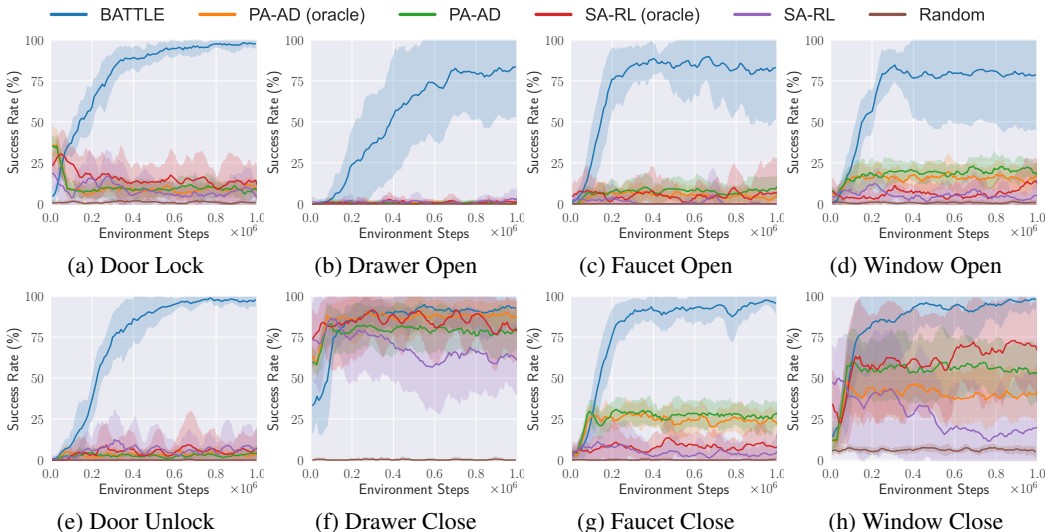

Figure 5: Training curves of all methods on various tasks in the opposite behaviors scenario. The solid line and shaded area denote the mean and the standard deviation of success rate over ten runs. In this scenario, the blue line (our method) outperforms all the baselines in both PbRL setting and oracle setting, which demonstrates the effectiveness of BATTLE.

et al., 2020; 2021; Sun et al., 2022), we consider using state attacks with $L^\infty$ norm in our experiments, and we report the mean and standard deviation across ten runs for all experiments. We also provide detailed hyper-parameter settings, implementation details and scenario design in Appendix F.

## 5.2 Manipulation on DRL Agents

We study the efficacy of our method compared to adversarial attack algorithms, which are adapted to our setting with minimal changes. Specifically, we devise two distinct scenarios on various simulated robotic manipulation tasks. Each victim agent is well-trained for a specific manipulation task.

**Scenarios on Manipulation.** In this scenario, we expect the robotic arm to reach a target coordinates instead of completing the original task. Figure 4 shows the training curves of baselines and our method on eight manipulation tasks. It shows that the performance of BATTLE surpasses that of the baselines by a large margin based on preference labels. To eliminate the influence of PbRL and further demonstrate the advantages of BATTLE, we additionally train the baseline methods with the ground-truth reward function and denote them as "oracle". We notice that the performance of SA-RL (oracle) greatly improves on several tasks over the preference-based version. However, BATTLE still outperforms SA-RL with oracle rewards on most tasks. These results demonstrate that BATTLE enables the agent to efficiently learn adversarial policy with human preferences. We also observe that PA-AD is incapable of mastering manipulation, even using the ground-truth rewards.

**Scenarios on Opposite Behaviors.** In the real world, robotic manipulation has good application values. Therefore, we design this scenario to quantitatively evaluate the vulnerability of these agents that masters various manipulation skills. Specifically, we expect each victim to complete the opposite task under the attack of the manipulator. For example, the victim which masters the skill of opening windows will close windows under targeted attack. As shown in Figure 5, BATTLE presents excellent performance and marginally shows obvious advantages over baseline methods on all tasks. The result again indicates that BATTLE is effective for a wide range of tasks and can efficiently learn adversarial policy with human preferences.

## 5.3 Manipulation on the Popular Offline RL Agents

In this experiment, we show the vulnerability of offline RL agents and demonstrate BATTLE can fool them into acting human desired behaviors. As for the implementation, we choose some online models[3] as victims, which are well-trained by official implementation with D4RL. We choose two tasks, Cheetah and Walker, using expert-level Decision Transformer agents as the victims. As shown

---

[3] https://huggingface.co/edbeeching

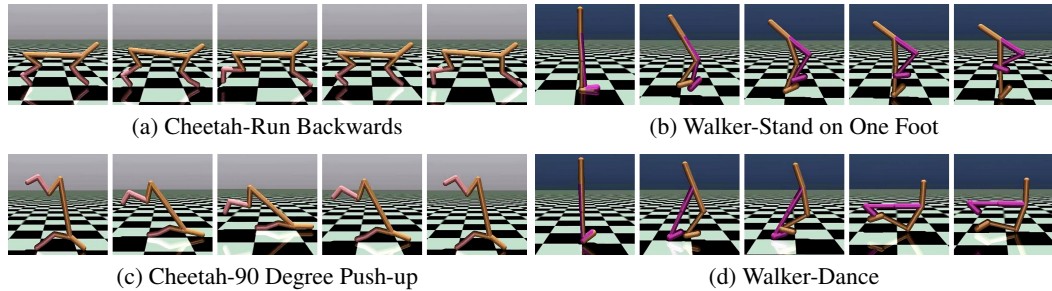

(a) Cheetah-Run Backwards
(b) Walker-Stand on One Foot

(c) Cheetah-90 Degree Push-up
(d) Walker-Dance

Figure 6: Human desired behaviors behaved by the Decision Transformer under the attack of BATTLE.

in Figure 6, Decision Transformer shows exploitable weaknesses and is misled to perform human desired behavior instead of the original task. Under the adversarial manipulation, the Cheetah agent runs backwards quickly in Figure 6a, and does 90 degree push-up in Figure 6c. The Walker agent stands on one foot for superior balance in Figure 6b, and dances with one leg lifted in Figure 6d. The results show that BATTLE can manipulate these victims to act behaviors consistent with human preferences and embodied agents are extremely vulnerable to these well-trained adversaries. We hope this experiment can inspire future work on the robustness of offline RL agents and embodied AI.

## 5.4    ROBUST AGENTS TRAINING AND EVALUATING

An intuitive application of BATTLE lies in evaluating the robustness of a given model or enhancing an agent's robustness through adversarial training. ATLA (Zhang et al., 2021) is a general training framework for robustness improvement, which alternately trains an agent and an adversary. Motivated by this, we introduce BATTLE-ATLA which trains an agent and a BATTLE attacker alternately. Table 1 shows that the performance of BATTLE-ATLA for a SAC agent, in comparison with state-of-the-art robust training methods. The experimental results summarize two aspects: firstly, BATTLE-ATLA significantly enhances the robustness of agents, and secondly, BATTLE can conduct stronger attacks on robust agents.

Table 1: Average episode rewards $\pm$ standard deviation of robust agents under different attack methods, and results are averaged across 100 episodes.

| Task | Model | BATTLE | PA-AD | SA-RL | Average Reward |
|------|-------|--------|-------|-------|----------------|
| Door Lock | BATTLE-ATLA | 874±444 | 628±486 | 503±120 | **668** |
| | PAAD-ATLA | 491±133 | 483±15 | 517±129 | 497 |
| | SARL-ATLA | 469±11 | 629±455 | 583±173 | 545 |
| Door Unlock | BATTLE-ATLA | 477±203 | 745±75 | 623±60 | **615** |
| | PAAD-ATLA | 398±12 | 381±11 | 398±79 | 389 |
| | SARL-ATLA | 393±36 | 377±8 | 385±26 | 385 |

## 5.5    ABLATION STUDY

**Contribution of Each Component.** We conduct additional experiments to investigate the effect of each component in BATTLE on Drawer Open, Drawer Close for the manipulation scenario and on Faucet Open, Faucet Close for the opposite behavior scenario. BATTLE contains three critical components: the weight function $h_\omega$, the intention policy $\pi_\theta$, and the combined policy. Table 2 shows that the intention policy plays an essential role in the BATTLE. As shown in Figure 7d, the intention policy can mitigate exploration difficulty caused by the restriction of victim policy and improve the exploration ability of BATTLE leading to a better adversary. We also observe that the combined policy balances the discrepancy between $\pi_\theta$ and $\pi_{\nu \circ \alpha}$ on the state distribution and improves the adversary's performance. In addition, we can economically train the weighting function to distinguish state importance by formulating the adversary learning as a bi-level optimization. It can further improve the asymptotic performance of BATTLE. These empirical results show that key ingredients of BATTLE are fruitfully wed and contribute to the BATTLE's success. To verify the restricted exploration problem, we visualize the exploration space of BATTLE and BATTLE without intention policy. Figure 7d shows that the intention policy significantly improve the exploration ability of BATTLE.

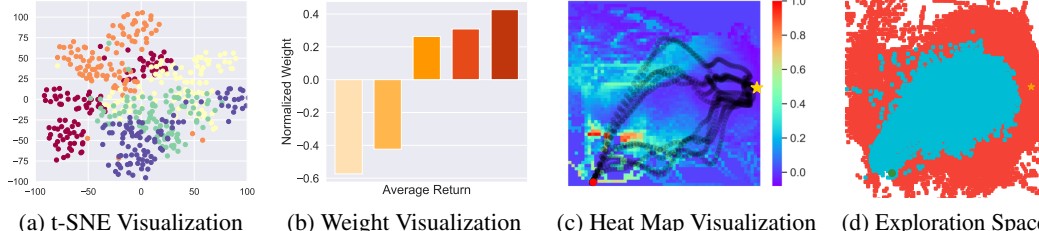

| (a) t-SNE Visualization | (b) Weight Visualization | (c) Heat Map Visualization | (d) Exploration Space |

Figure 7: (a) A visualization of the weights of trajectories of different qualities by five different policies. (b) Trajectory weights generated by the weighting function from different policies are extracted and visualized with t-SNE. (c) A heat map showing the weight distribution and the trajectory of the perturbed agent in 2D coordinates. The red point denotes the start position and the yellow star indicates the targeted position. (d) A visualization of the exploration space of BATTLE (red) and BATTLE without intention policy (blue). The green point denotes the start and the yellow star denotes the target position.

Table 2: Effects of each component. The success rate on four simulated robotic manipulation tasks from Meta-world. The results are the average success rate across five runs.

| Task Type | Task | BATTLE | BATTLE w/o $h_\omega$ | BATTLE w/o $\pi_\theta$ | BATTLE w/o combination |
|---|---|---|---|---|---|
| Manipulation | Drawer Open | **99.1%** | 91.3% | 21.7% | 68.0% |
| | Drawer Close | **80.9%** | 70.2% | 8.0% | 26.0% |
| Opposite | Faucet Open | 84.4% | **89.8%** | 0.0% | 57.0% |
| | Faucet Close | **95.1%** | 94.1% | 13.0% | 59.1% |

**Effects of the Weighting Function.** To further understand the weighting function proposed in Section 4, we conduct experimental data analysis and visualization from multiple perspectives. Five perturbed policies are uniformly sampled with performance increase sequentially before BATTLE convergence. For each policy, we roll out 100 trajectories and obtain the trajectory weight vectors via the weighting function. By leveraging the technique of t-SNE (van der Maaten & Hinton, 2008), the weight vectors of different policies are visualized in Figure 7a. From the figure, we can observe clear boundaries between the trajectory weights of various policies, suggesting that the weighting function can distinguish trajectories of different qualities. In Figure 7b, the darker color indicates trajectories with higher success rates of manipulation. The result shows that the weighting function gives higher weights to better trajectories for improving the adversarial policy performance. To further illustrate the effect of the weighting function, we present a heat map of the weight distribution in 2D coordinates and annotate part of the trajectories of the perturbed policy. As Figure 7c shows, the weighting function scores the surrounding states in trajectories from the perturbed policy higher, especially in the early stage before reaching the target point.

Extensive experiments are conducted to analyze and discuss the impact of feedback amount, attack budgets on the performance of BATTLE and quality of learned reward functions in the Appendix G.

## 6 CONCLUSION

In this paper, we propose BATTLE, a behavior-oriented adversarial attack approach against DRL learners, which can manipulate the victim to perform desired behaviors of human. BATTLE involves an adversary adding imperceptible perturbations on the observations of the victim, an intention policy learned through PbRL as a flexible behavior orientation, and a weighting function to identify essential states for the efficient adversarial attack. We analyze the convergence of BATTLE and prove that BATTLE converges to critical points under some mild conditions. Empirically, we design two scenarios on several manipulation tasks of Meta-world, and the results demonstrate that BATTLE outperforms the baselines under the targeted adversarial setting. Additionally, BATTLE can enhance the robustness of agents by training with adversary. We further show embodied agents' vulnerability by attacking Decision Transformer on some MuJoCo tasks.

ETHICS STATEMENT

Preference-based RL provides an effective way to train agents without a carefully designed reward function. However, learning from human preferences means humans need to provide labeled data which inevitably has biases introducing systematic error. While there are possible negative impacts when malicious people attack other policies using our methods. However, our approach also makes other researchers aware of the vulnerability of policies for AI safety. Furthermore, our method might serve as a red teaming tool to evaluate DRL agents for potential unsafe behaviors.

REPRODUCIBILITY STATEMENT

The details of experiment settings are provided in Section 4. We provide detailed proofs of theoretical analysis in Appendix D. A more detailed description and implementation setting can be found in Appendix F. Meanwhile, we present the link of our videos in the abstract and we will provide source code during rebuttal.

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

# A THE FULL PROCEDURE OF BATTLE

**The Combined Policy.** In order to address the inefficiency caused by the discrepancy between $\pi_\theta$ and $\pi_{\nu \circ \alpha}$ in the state distribution, we propose a strategy to construct the behavior policy $\pi$ for data collection in our practical implementation. Inspired by Branched rollout (Janner et al., 2019), we combine the intention policy $\pi_\theta$ with the perturbed policy $\pi_{\nu \circ \alpha}$. Specifically, we define $\pi^{1:h} = \pi_{\nu \circ \alpha}^{1:h}$ and $\pi^{h+1:H} = \pi_\theta^{h+1:H}$, where $h$ is sampled from a uniform distribution $U(0, H)$ and $H$ represents the task horizon. The resulting combined policy $\pi$ is responsible for data collection, which is then stored in the replay buffer during the learning process.

We present the detailed procedures of our proposed method in Algorithm 1. Our method, referred to as BATTLE, is built upon the well-established preference-based RL algorithm PEBBLE (Lee et al., 2021a).

---

**Algorithm 1** BATTLE

**Input:** a fixed victim policy $\pi_\nu$, frequency of human feedback $K$, outer loss updating frequency $M$, task horizon $H$
1: Initialize parameters of $Q_\phi$, $\pi_\theta$, $\widehat{r}_\psi$, $\pi_\alpha$ and $h_\omega$
2: Initialize $\mathcal{B}$ and $\pi_\theta$ with unsupervised exploration
3: Initialize preference data set $\mathcal{D} \leftarrow \emptyset$
4: **for** each iteration **do**
5:     // Construct the combined policy $\pi$
6:     **if** episode is done **then**
7:         $h \sim U(0, H)$
8:         $\pi^{1:h} = \pi_{\nu \circ \alpha}^{1:h}$ and $\pi^{h+1:H} = \pi_\theta^{h+1:H}$
9:     **end if**
10:     Take action $a_t \sim \pi$ and collect $s_{t+1}$
11:     Store transition into dataset $\mathcal{B} \leftarrow \mathcal{B} \cup \{(s_t, a_t, \widehat{r}_\psi(s_t), s_{t+1})\}$
12:     // Query preference and Reward learning
13:     **if** iteration % $K == 0$ **then**
14:         **for** each query step **do**
15:             Sample pair of trajectories $(\sigma^0, \sigma^1)$
16:             Query preference $y$ from manipulator
17:             Store preference data into dataset $\mathcal{D} \leftarrow \mathcal{D} \cup \{(\sigma^0, \sigma^1, y)\}$
18:         **end for**
19:         **for** each gradient step **do**
20:             Sample batch $\{(\sigma^0, \sigma^1, y)_i\}_{i=1}^n$ from $\mathcal{D}$
21:             Optimize (2) to update $\widehat{r}_\psi$
22:         **end for**
23:     **end if**
24:     // Inner loss optimization
25:     **for** each gradient step **do**
26:         Sample random mini-batch transitions from $\mathcal{B}$
27:         Optimize $\pi_\alpha$: minimize (6) with respect to $\alpha$
28:     **end for**
29:     // Outer loss optimization
30:     **if** iteration % $M == 0$ **then**
31:         Sample random mini-batch transitions from $\mathcal{B}$
32:         Optimize $h_\omega$: minimize (7) with respect to $\omega$
33:     **end if**
34:     // Intention policy learning
35:     Update $Q_\phi$ and $\pi_\theta$ according to (3) and (4), respectively.
36: **end for**
**Output:** adversarial policy $\pi_\alpha$

---

## B   DERIVATION OF THE GRADIENT OF THE OUTER-LEVEL LOSS

In this section, we present detailed derivation of the gradient of the outer loss $J_\pi$ with respect to the parameters of the weighting function $\omega$. According to the chain rule, we can derive that

$$
\begin{aligned}
&\nabla_\omega J_\pi(\hat\alpha(\omega))\big|_{\omega_t} \\
&= \frac{\partial J_\pi(\hat\alpha(\omega))}{\partial \hat\alpha(\omega)}\bigg|_{\hat\alpha_t} \frac{\partial \hat\alpha_t(\omega)}{\partial \omega}\bigg|_{\omega_t} \\
&= \frac{\partial J_\pi(\hat\alpha(\omega))}{\partial \hat\alpha(\omega)}\bigg|_{\hat\alpha_t} \frac{\partial \hat\alpha_t(\omega)}{\partial h(\mathbf{s};\omega)}\bigg|_{\omega_t} \frac{\partial h(\mathbf{s};\omega)}{\partial \omega}\bigg|_{\omega_t} \\
&= -\eta_t \frac{\partial J_\pi(\hat\alpha(\omega))}{\partial \hat\alpha(\omega)}\bigg|_{\hat\alpha_t} \sum_{\mathbf{s}\sim\mathcal{B}} \frac{\partial D_{\mathrm{KL}}\left(\pi_{\nu\circ\alpha}(\mathbf{s}) \,\|\, \pi_\theta(\mathbf{s})\right)}{\partial \alpha}\bigg|_{\alpha_t} \frac{\partial h(\mathbf{s};\omega)}{\partial \omega}\bigg|_{\omega_t} \\
&= -\eta_t \sum_{\mathbf{s}\sim\mathcal{B}} \left( \frac{\partial J_\pi(\hat\alpha(\omega))}{\partial \hat\alpha(\omega)}\bigg|_{\hat\alpha_t}^\top \frac{\partial D_{\mathrm{KL}}\left(\pi_{\nu\circ\alpha}(\mathbf{s}) \,\|\, \pi_\theta(\mathbf{s})\right)}{\partial \alpha}\bigg|_{\alpha_t} \right) \frac{\partial h(\mathbf{s};\omega)}{\partial \omega}\bigg|_{\omega_t}.
\end{aligned}
\tag{10}
$$

For brevity of expression, we let:

$$
f(\mathbf{s}) = \frac{\partial J_\pi(\hat\alpha(\omega))}{\partial \hat\alpha(\omega)}\bigg|_{\hat\alpha_t}^\top \frac{\partial D_{\mathrm{KL}}\left(\pi_{\nu\circ\alpha}(\mathbf{s}) \,\|\, \pi_\theta(\mathbf{s})\right)}{\partial \hat\alpha}\bigg|_{\alpha_t}.
\tag{11}
$$

The gradient of outer-level optimization loss with respect to parameters $\omega$ is:

$$
\nabla_\omega J_\pi(\hat\alpha(\omega))\big|_{\omega_t} = -\eta_t \sum_{\mathbf{s}\sim\mathcal{B}} f(\mathbf{s}) \cdot \frac{\partial h(\mathbf{s};\omega)}{\partial \omega}\bigg|_{\omega_t}.
\tag{12}
$$

## C   CONNECTION BETWEEN RSA-MDP AND MDP

**Lemma C.1.** *Given a RSA-MDP $\mathcal{M} = (\mathcal{S}, \mathcal{A}, \mathcal{B}, \widehat{\mathcal{R}}, \mathcal{P}, \gamma)$ and a fixed victim policy $\pi_\nu$, there exists a MDP $\hat{\mathcal{M}} = (\mathcal{S}, \hat{\mathcal{A}}, \widehat{\mathcal{R}}, \widehat{\mathcal{P}}, \gamma)$ such that the optimal policy of $\hat{\mathcal{M}}$ is equivalent to the optimal adversary $\pi_\alpha$ in RSA-MDP given a fixed victim, where $\widehat{\mathcal{A}} = \mathcal{S}$ and*

$$
\widehat{\mathcal{P}}(\mathbf{s}'|\mathbf{s}, \mathbf{a}) = \sum_{\mathbf{a}\in\mathcal{A}} \pi_\nu(\mathbf{a}|\widehat{\mathbf{a}})\mathcal{P}(\mathbf{s}'|\mathbf{s}, \mathbf{a}) \quad \text{for } \mathbf{s}, \mathbf{s}' \in \mathcal{S} \text{ and } \widehat{\mathbf{a}} \in \widehat{\mathcal{A}}.
$$

## D   THEORETICAL ANALYSIS AND PROOFS

### D.1   THEOREM 1: CONVERGENCE RATE OF THE OUTER LOSS

**Lemma D.1.** *(Lemma 1.2.3 in Nesterov (1998)) If function $f(x)$ is Lipschitz smooth on $\mathbb{R}^n$ with constant L, then $\forall x, y \in \mathbb{R}^n$, we have*

$$
\left| f(y) - f(x) - f'(x)^\top (y - x) \right| \le \frac{L}{2} \|y - x\|^2.
\tag{13}
$$

*Proof.* $\forall x, y \in \mathbb{R}^n$, we have

$$
\begin{aligned}
f(y) &= f(x) + \int_0^1 f'(x + \tau(y - x))^\top (y - x) d\tau \\
&= f(x) + f'(x)^\top (y - x) + \int_0^1 [f'(x + \tau(y - x)) - f'(x)]^\top (y - x) d\tau.
\end{aligned}
\tag{14}
$$

Then we can derive that

$$
\begin{aligned}
\left| f(y) - f(x) - f'(x)^\top (y - x) \right| &= \left| \int_0^1 [f'(x + \tau(y - x)) - f'(x)]^\top (y - x) d\tau \right| \\
&\le \int_0^1 \left| [f'(x + \tau(y - x)) - f'(x)]^\top (y - x) \right| d\tau \\
&\le \int_0^1 \|f'(x + \tau(y - x)) - f'(x)\| \cdot \|y - x\| d\tau \\
&\le \int_0^1 \tau L \|y - x\|^2 d\tau = \frac{L}{2} \|y - x\|^2,
\end{aligned}
\tag{15}
$$

where the first inequality holds for $\left| \int_a^b f(x)dx \right| \leq \int_a^b |f(x)| \, dx$, the second inequality holds for Cauchy-Schwarz inequality, and the last inequality holds for the definition of Lipschitz smoothness. $\qquad \square$

**Theorem D.2.** *Suppose $J_\pi$ is Lipschitz-smooth with constant $L$, the gradient of $J_\pi$ and $\mathcal{L}_{att}$ is bounded by $\rho$. Let the training iterations be $T$, the inner-level optimization learning rate $\eta_t = \min\{1, \frac{c_1}{T}\}$ for some constant $c_1 > 0$ where $\frac{c_1}{T} < 1$. Let the outer-level optimization learning rate $\beta_t = \min\{\frac{1}{L}, \frac{c_2}{\sqrt{T}}\}$ for some constant $c_2 > 0$ where $c_2 \leq \frac{\sqrt{T}}{L}$, and $\sum_{t=1}^\infty \beta_t \leq \infty, \sum_{t=1}^\infty \beta_t^2 \leq \infty$. The convergence rate of $J_\pi$ achieves*

$$\min_{1 \leq t \leq T} \mathbb{E}\left[\|\nabla_\omega J_\pi(\alpha_{t+1}(\omega_t))\|^2\right] \leq \mathcal{O}\left(\frac{1}{\sqrt{T}}\right). \tag{16}$$

*Proof.* First,

$$
\begin{aligned}
& J_\pi(\hat{\alpha}_{t+2}(\omega_{t+1})) - J_\pi(\hat{\alpha}_{t+1}(\omega_t)) \\
& = \{J_\pi(\hat{\alpha}_{t+2}(\omega_{t+1})) - J_\pi(\hat{\alpha}_{t+1}(\omega_{t+1}))\} + \{J_\pi(\hat{\alpha}_{t+1}(\omega_{t+1})) - J_\pi(\hat{\alpha}_{t+1}(\omega_t))\}.
\end{aligned}
\tag{17}
$$

Then we separately derive the two terms of (17). For the first term,

$$
\begin{aligned}
& J_\pi(\hat{\alpha}_{t+2}(\omega_{t+1})) - J_\pi(\hat{\alpha}_{t+1}(\omega_{t+1})) \\
& \leq \nabla_{\hat{\alpha}} J_\pi(\hat{\alpha}_{t+1}(\omega_{t+1}))^\top (\hat{\alpha}_{t+2}(\omega_{t+1}) - \hat{\alpha}_{t+1}(\omega_{t+1})) + \frac{L}{2}\|\hat{\alpha}_{t+2}(\omega_{t+1}) - \hat{\alpha}_{t+1}(\omega_{t+1})\|^2 \\
& \leq \|\nabla_{\hat{\alpha}} J_\pi(\hat{\alpha}_{t+1}(\omega_{t+1}))\| \cdot \|\hat{\alpha}_{t+2}(\omega_{t+1}) - \hat{\alpha}_{t+1}(\omega_{t+1})\| + \frac{L}{2}\|\hat{\alpha}_{t+2}(\omega_{t+1}) - \hat{\alpha}_{t+1}(\omega_{t+1})\|^2 \\
& \leq \rho \cdot \|-\eta_{t+1}\nabla_{\hat{\alpha}}\mathcal{L}_{att}(\hat{\alpha}_{t+1})\| + \frac{L}{2}\|-\eta_{t+1}\nabla_{\hat{\alpha}}\mathcal{L}_{att}(\hat{\alpha}_{t+1})\|^2 \\
& \leq \eta_{t+1}\rho^2 + \frac{L}{2}\eta_{t+1}^2\rho^2,
\end{aligned}
\tag{18}
$$

where $\hat{\alpha}_{t+2}(\omega_{t+1}) - \hat{\alpha}_{t+1}(\omega_{t+1}) = -\eta_{t+1}\nabla_{\hat{\alpha}}\mathcal{L}_{att}(\hat{\alpha}_{t+1})$, the first inequality holds for Lemma D.1, the second inequality holds for Cauchy-Schwarz inequality, the third inequality holds for $\|\nabla_{\hat{\alpha}} J_\pi(\hat{\alpha}_{t+1}(\omega_{t+1}))\| \leq \rho$, and the last inequality holds for $\|\nabla_{\hat{\alpha}}\mathcal{L}_{att}(\hat{\alpha}_{t+1})\| \leq \rho$. It can be proved that the gradient of $\omega$ with respect to $J_\pi$ is Lipschitz continuous and we assume the Lipschitz constant is $L$. Therefore, for the second term,

$$
\begin{aligned}
& J_\pi(\hat{\alpha}_{t+1}(\omega_{t+1})) - J_\pi(\hat{\alpha}_{t+1}(\omega_t)) \\
& \leq \nabla_\omega J_\pi(\hat{\alpha}_{t+1}(\omega_t))^\top (\omega_{t+1} - \omega_t) + \frac{L}{2}\|\omega_{t+1} - \omega_t\|^2 \\
& = -\beta_t \nabla_\omega J_\pi(\hat{\alpha}_{t+1}(\omega_t))^\top \nabla_\omega J_\pi(\hat{\alpha}_{t+1}(\omega_t)) + \frac{L\beta_t^2}{2}\|\nabla_\omega J_\pi(\hat{\alpha}_{t+1}(\omega_t))\|^2 \\
& = -(\beta_t - \frac{L\beta_t^2}{2})\|\nabla_\omega J_\pi(\hat{\alpha}_{t+1}(\omega_t))\|^2,
\end{aligned}
\tag{19}
$$

where $\omega_{t+1} - \omega_t = -\beta_t \nabla_\omega J_\pi(\hat{\alpha}_{t+1}(\omega_t))$, and the first inequality holds for Lemma D.1. Therefore, (17) becomes

$$
J_\pi(\hat{\alpha}_{t+2}(\omega_{t+1})) - J_\pi(\hat{\alpha}_{t+1}(\omega_t)) \leq \eta_{t+1}\rho^2 + \frac{L}{2}\eta_{t+1}^2\rho^2 - (\beta_t - \frac{L\beta_t^2}{2})\|\nabla_\omega J_\pi(\hat{\alpha}_{t+1}(\omega_t))\|^2.
\tag{20}
$$

Rearranging the terms of (20), we obtain

$$
(\beta_t - \frac{L\beta_t^2}{2})\|\nabla_\omega J_\pi(\hat{\alpha}_{t+1}(\omega_t))\|^2 \leq J_\pi(\hat{\alpha}_{t+1}(\omega_t)) - J_\pi(\hat{\alpha}_{t+2}(\omega_{t+1})) + \eta_{t+1}\rho^2 + \frac{L}{2}\eta_{t+1}^2\rho^2.
\tag{21}
$$

Then, we sum up both sides of (21),

$$\sum_{t=1}^{T}(\beta_t - \frac{L\beta_t^2}{2})\left\|\nabla_\omega J_\pi(\hat{\alpha}_{t+1}(\omega_t))\right\|^2$$

$$\leq J_\pi(\hat{\alpha}_2(\omega_1)) - J_\pi(\hat{\alpha}_{T+2}(\omega_{T+1})) + \sum_{t=1}^{T}(\eta_{t+1}\rho^2 + \frac{L}{2}\eta_{t+1}^2\rho^2) \tag{22}$$

$$\leq J_\pi(\hat{\alpha}_2(\omega_1)) + \sum_{t=1}^{T}(\eta_{t+1}\rho^2 + \frac{L}{2}\eta_{t+1}^2\rho^2).$$

Therefore,

$$\min_{1\leq t\leq T}\mathbb{E}\left[\left\|\nabla_\omega J_\pi(\hat{\alpha}_{t+1}(\omega_t))\right\|^2\right]$$

$$\leq \frac{\sum_{t=1}^{T}(\beta_t - \frac{L\beta_t^2}{2})\left\|\nabla_\omega J_\pi(\hat{\alpha}_{t+1}(\omega_t))\right\|^2}{\sum_{t=1}^{T}(\beta_t - \frac{L\beta_t^2}{2})}$$

$$\leq \frac{1}{\sum_{t=1}^{T}(2\beta_t - L\beta_t^2)}\left[2J_\pi(\hat{\alpha}_2(\omega_1)) + \sum_{t=1}^{T}(2\eta_{t+1}\rho^2 + L\eta_{t+1}^2\rho^2)\right]$$

$$\leq \frac{1}{\sum_{t=1}^{T}\beta_t}\left[2J_\pi(\hat{\alpha}_2(\omega_1)) + \sum_{t=1}^{T}\eta_{t+1}\rho^2(2 + L\eta_{t+1})\right]$$

$$\leq \frac{1}{T\beta_t}\left[2J_\pi(\hat{\alpha}_2(\omega_1)) + T\eta_{t+1}\rho^2(2 + L)\right] \tag{23}$$

$$= \frac{2J_\pi(\hat{\alpha}_2(\omega_1))}{T\beta_t} + \frac{\eta_{t+1}\rho^2(2 + L)}{\beta_t}$$

$$= \frac{2J_\pi(\hat{\alpha}_2(\omega_1))}{T}\max\{L, \frac{\sqrt{T}}{c_2}\} + \min\{1, \frac{c_1}{T}\}\max\{L, \frac{\sqrt{T}}{c_2}\}\rho^2(2 + L)$$

$$\leq \frac{2J_\pi(\hat{\alpha}_2(\omega_1))}{c_2\sqrt{T}} + \frac{c_1\rho^2(2 + L)}{c_2\sqrt{T}}$$

$$= \mathcal{O}\left(\frac{1}{\sqrt{T}}\right),$$

where the second inequality holds according to (22), the third inequality holds for $\sum_{t=1}^{T}\left(2\beta_t - L\beta_t^2\right) \geq \sum_{t=1}^{T}\beta_t$. $\qquad\square$

## D.2 THEOREM 2: CONVERGENCE OF THE INNER LOSS

**Lemma D.3.** *(Lemma A.5 in Mairal (2013)) Let $(a_n)_{n\geq 1}, (b_n)_{n\geq 1}$ be two non-negative real sequences such that the series $\sum_{n=1}^{\infty} a_n$ diverges, the series $\sum_{n=1}^{\infty} a_n b_n$ converges, and there exists $C > 0$ such that $|b_{n+1} - b_n| \leq Ca_n$. Then, the sequence $(b_n)_{n\geq 1}$ converges to 0.*

**Theorem D.4.** *Suppose $J_\pi$ is Lipschitz-smooth with constant L, the gradient of $J_\pi$ and $\mathcal{L}_{att}$ is bounded by $\rho$. Let the training iterations be T, the inner-level optimization learning rate $\eta_t = \min\{1, \frac{c_1}{T}\}$ for some constant $c_1 > 0$ where $\frac{c_1}{T} < 1$. Let the outer-level optimization learning rate $\beta_t = \min\{\frac{1}{L}, \frac{c_2}{\sqrt{T}}\}$ for some constant $c_2 > 0$ where $c_2 \leq \frac{\sqrt{T}}{L}$, and $\sum_{t=1}^{\infty}\beta_t \leq \infty, \sum_{t=1}^{\infty}\beta_t^2 \leq \infty$. $\mathcal{L}_{att}$ achieves*

$$\lim_{t\to\infty}\mathbb{E}\left[\left\|\nabla_\alpha\mathcal{L}_{att}(\alpha_t; \omega_t)\right\|^2\right] = 0. \tag{24}$$

*Proof.* First,

$$\mathcal{L}_{att}(\alpha_{t+1}; \omega_{t+1}) - \mathcal{L}_{att}(\alpha_t; \omega_t)$$
$$= \{\mathcal{L}_{att}(\alpha_{t+1}; \omega_{t+1}) - \mathcal{L}_{att}(\alpha_{t+1}; \omega_t)\} + \{\mathcal{L}_{att}(\alpha_{t+1}; \omega_t) - \mathcal{L}_{att}(\alpha_t; \omega_t)\}. \tag{25}$$

For the first term in (25),

$$
\begin{aligned}
&\mathcal{L}_{\text{att}}(\alpha_{t+1}; \omega_{t+1}) - \mathcal{L}_{\text{att}}(\alpha_{t+1}; \omega_t) \\
&\leq \nabla_\omega \mathcal{L}_{\text{att}}(\alpha_{t+1}; \omega_t)^\top (\omega_{t+1} - \omega_t) + \frac{L}{2} \|\omega_{t+1} - \omega_t\|^2 \\
&= -\beta_t \nabla_\omega \mathcal{L}_{\text{att}}(\alpha_{t+1}; \omega_t)^\top \nabla_\omega J_\pi(\alpha_{t+1}(\omega_t)) + \frac{L\beta_t^2}{2} \|\nabla_\omega J_\pi(\alpha_{t+1}(\omega_t))\|^2 .
\end{aligned}
\tag{26}
$$

where $\omega_{t+1} - \omega_t = -\beta_t \nabla_\omega J_\pi(\alpha_{t+1}(\omega_t))$, and the first inequality holds according to Lemma D.1.
For the second term in (25),

$$
\begin{aligned}
&\mathcal{L}_{\text{att}}(\alpha_{t+1}; \omega_t) - \mathcal{L}_{\text{att}}(\alpha_t; \omega_t) \\
&\leq \nabla_\alpha \mathcal{L}_{\text{att}}(\alpha_t; \omega_t)^\top (\alpha_{t+1} - \alpha_t) + \frac{L}{2} \|\alpha_{t+1} - \alpha_t\|^2 \\
&= -\eta_t \nabla_\alpha \mathcal{L}_{\text{att}}(\alpha_t; \omega_t)^\top \nabla_\alpha \mathcal{L}_{\text{att}}(\alpha_t; \omega_t) + \frac{L\eta_t^2}{2} \|\nabla_\alpha \mathcal{L}_{\text{att}}(\alpha_t; \omega_t)\|^2 \\
&= -(\eta_t - \frac{L\eta_t^2}{2}) \|\nabla_\alpha \mathcal{L}_{\text{att}}(\alpha_t; \omega_t)\|^2 .
\end{aligned}
\tag{27}
$$

where $\alpha_{t+1} - \alpha_t = -\eta_t \nabla_\alpha \mathcal{L}_{\text{att}}(\alpha_t; \omega_t)$, and the first inequality holds according to Lemma (D.1).
Therefore, (25) becomes

$$
\begin{aligned}
&\mathcal{L}_{\text{att}}(\alpha_{t+1}; \omega_{t+1}) - \mathcal{L}_{\text{att}}(\alpha_t; \omega_t) \\
&\leq -\beta_t \nabla_\omega \mathcal{L}_{\text{att}}(\alpha_{t+1}; \omega_t)^\top \nabla_\omega J_\pi(\alpha_{t+1}(\omega_t)) + \frac{L\beta_t^2}{2} \|\nabla_\omega J_\pi(\alpha_{t+1}(\omega_t))\|^2 \\
&\quad -(\eta_t - \frac{L\eta_t^2}{2}) \|\nabla_\alpha \mathcal{L}_{\text{att}}(\alpha_t; \omega_t)\|^2 .
\end{aligned}
\tag{28}
$$

Taking expectation of both sides of (28) and rearranging the terms, we obtain

$$
\begin{aligned}
&\eta_t \mathbb{E}\left[\|\nabla_\alpha \mathcal{L}_{\text{att}}(\alpha_t; \omega_t)\|^2\right] + \beta_t \mathbb{E}\left[\|\nabla_\omega \mathcal{L}_{\text{att}}(\alpha_{t+1}; \omega_t)\| \cdot \|\nabla_\omega J_\pi(\alpha_{t+1}(\omega_t))\|\right] \\
&\leq \mathbb{E}\left[\mathcal{L}_{\text{att}}(\alpha_t; \omega_t)\right] - \mathbb{E}\left[\mathcal{L}_{\text{att}}(\alpha_{t+1}; \omega_{t+1})\right] + \frac{L\beta_t^2}{2} \mathbb{E}\left[\|\nabla_\omega J_\pi(\alpha_{t+1}(\omega_t))\|^2\right] \\
&\quad + \frac{L\eta_t^2}{2} \mathbb{E}\left[\|\nabla_\alpha \mathcal{L}_{\text{att}}(\alpha_t; \omega_t)\|^2\right] .
\end{aligned}
\tag{29}
$$

Summing up both sides of (29) from $t = 1$ to $\infty$,

$$
\begin{aligned}
&\sum_{t=1}^\infty \eta_t \mathbb{E}\left[\|\nabla_\alpha \mathcal{L}_{\text{att}}(\alpha_t; \omega_t)\|^2\right] + \sum_{t=1}^\infty \beta_t \mathbb{E}\left[\|\nabla_\omega \mathcal{L}_{\text{att}}(\alpha_{t+1}; \omega_t)\| \cdot \|\nabla_\omega J_\pi(\alpha_{t+1}(\omega_t))\|\right] \\
&\leq \mathbb{E}\left[\mathcal{L}_{\text{att}}(\alpha_1; \omega_1)\right] - \lim_{t\to\infty} \mathbb{E}\left[\mathcal{L}_{\text{att}}(\alpha_{t+1}; \omega_{t+1})\right] + \sum_{t=1}^\infty \frac{L\beta_t^2}{2} \mathbb{E}\left[\|\nabla_\omega J_\pi(\alpha_{t+1}(\omega_t))\|^2\right] \\
&\quad + \sum_{t=1}^\infty \frac{L\eta_t^2}{2} \mathbb{E}\left[\|\nabla_\alpha \mathcal{L}_{\text{att}}(\alpha_t; \omega_t)\|^2\right] \\
&\leq \sum_{t=1}^\infty \frac{L(\eta_t^2 + \beta_t^2)\rho^2}{2} + \mathbb{E}\left[\mathcal{L}_{\text{att}}(\alpha_1; \omega_1)\right] \leq \infty,
\end{aligned}
\tag{30}
$$

where the second inequality holds for $\sum_{t=1}^\infty \eta_t^2 \leq \infty$, $\sum_{t=1}^\infty \beta_t^2 \leq \infty$, $\|\nabla_\alpha \mathcal{L}_{\text{att}}(\alpha_t; \omega_t)\| \leq \rho$, $\|\nabla_\omega J_\pi(\alpha_{t+1}(\omega_t))\| \leq \rho$. Since

$$
\sum_{t=1}^\infty \beta_t \mathbb{E}\left[\|\nabla_\omega \mathcal{L}_{\text{att}}(\alpha_{t+1}; \omega_t)\| \cdot \|\nabla_\omega J_\pi(\alpha_{t+1}(\omega_t))\|\right] \leq L\rho \sum_{t=1}^\infty \beta_t \leq \infty.
\tag{31}
$$

Therefore, we have

$$
\sum_{t=1}^\infty \eta_t \mathbb{E}\left[\|\nabla_\alpha \mathcal{L}_{\text{att}}(\alpha_t; \omega_t)\|^2\right] < \infty.
\tag{32}
$$

Since $|(\|a\| + \|b\|)(\|a\| - \|b\|)| \leq \|a + b\|\|a - b\|$, we can derive that

$$
\begin{aligned}
&\left| \mathbb{E}\left[ \|\nabla_\alpha \mathcal{L}_{\text{att}}(\alpha_{t+1}; \omega_{t+1})\|^2 \right] - \mathbb{E}\left[ \|\nabla_\alpha \mathcal{L}_{\text{att}}(\alpha_t; \omega_t)\|^2 \right] \right| \\
=& \left| \mathbb{E}\left[ \left( \|\nabla_\alpha \mathcal{L}_{\text{att}}(\alpha_{t+1}; \omega_{t+1})\| + \|\nabla_\alpha \mathcal{L}_{\text{att}}(\alpha_t; \omega_t)\| \right) + \left( \|\nabla_\alpha \mathcal{L}_{\text{att}}(\alpha_{t+1}; \omega_{t+1})\| - \|\nabla_\alpha \mathcal{L}_{\text{att}}(\alpha_t; \omega_t)\| \right) \right] \right| \\
\leq& \mathbb{E}\left[ \left| \|\nabla_\alpha \mathcal{L}_{\text{att}}(\alpha_{t+1}; \omega_{t+1})\| + \|\nabla_\alpha \mathcal{L}_{\text{att}}(\alpha_t; \omega_t)\| \right| \left| \|\nabla_\alpha \mathcal{L}_{\text{att}}(\alpha_{t+1}; \omega_{t+1})\| - \|\nabla_\alpha \mathcal{L}_{\text{att}}(\alpha_t; \omega_t)\| \right| \right] \\
\leq& \mathbb{E}\left[ \|\nabla_\alpha \mathcal{L}_{\text{att}}(\alpha_{t+1}; \omega_{t+1}) + \nabla_\alpha \mathcal{L}_{\text{att}}(\alpha_t; \omega_t)\| \cdot \|\nabla_\alpha \mathcal{L}_{\text{att}}(\alpha_{t+1}; \omega_{t+1}) - \nabla_\alpha \mathcal{L}_{\text{att}}(\alpha_t; \omega_t)\| \right] \\
\leq& \mathbb{E}\left[ \left( \|\nabla_\alpha \mathcal{L}_{\text{att}}(\alpha_{t+1}; \omega_{t+1})\| + \|\nabla_\alpha \mathcal{L}_{\text{att}}(\alpha_t; \omega_t)\| \right) \|\nabla_\alpha \mathcal{L}_{\text{att}}(\alpha_{t+1}; \omega_{t+1}) - \nabla_\alpha \mathcal{L}_{\text{att}}(\alpha_t; \omega_t)\| \right] \\
\leq& 2L\rho \mathbb{E}\left[ \|(\alpha_{t+1}, \omega_{t+1}) - (\alpha_t, \omega_t)\| \right] \\
\leq& 2L\rho \eta_t \beta_t \mathbb{E}\left[ \|(\nabla_\alpha \mathcal{L}_{\text{att}}(\alpha_t; \omega_t), \nabla_\omega J_\pi(\alpha_{t+1}(\omega_t)))\| \right] \\
\leq& 2L\rho \eta_t \beta_t \sqrt{ \mathbb{E}\left[ \|\nabla_\alpha \mathcal{L}_{\text{att}}(\alpha_t; \omega_t)\|^2 \right] + \mathbb{E}\left[ \|\nabla_\omega J_\pi(\alpha_{t+1}(\omega_t))\|^2 \right] } \\
\leq& 2L\rho \eta_t \beta_t \sqrt{2\rho^2} \\
\leq& 2\sqrt{2} L\rho^2 \eta_t \beta_t.
\end{aligned}
$$
(33)

Since $\sum_{t=1}^{\infty} \eta_t = \infty$, according to Lemma D.3, we have

$$
\lim_{t \to \infty} \mathbb{E}\left[ \|\nabla_\alpha \mathcal{L}_{\text{att}}(\alpha_t; \omega_t)\|^2 \right] = 0.
$$
(34)

$\square$

# E    DETAILS OF PBRL

In this section, we present details of the scripted teacher and preference collection. It is a crucial part of the PbRL, and BATTLE follows these settings as Lee et al. (2021a).

**Scripted Teacher.** To evaluate the performance systemically, a useful way is to consider a scripted teacher that provides preferences between a pair of agent's trajectory segments according to the oracle reward function. Leveraging the preference labels from the human teacher is ideal, while it is hard to evaluate algorithms quantitatively and quickly. Specifically, the scripted teacher can immediately provide ground truth rewards based on the state $\mathbf{s}$ and action $\mathbf{a}$. It is a function designed to approximate the human's intention.

**Preference Collection.** During training, we need to query human preference labels at regular intervals. It samples a batch of segment pairs and calculates the cumulative reward of each segment with rewards provided by the scripted teacher. For a specific segment pair, human prefers the segment with a larger cumulative reward. The segment with a larger cumulative reward is labelled with 1, and the smaller one is labelled with 0. As for the computational cost, we assume that $M$ preference labels are required, the segment length is $N$ in a run, and the time complexity is $\mathcal{O}(MN)$. However, it is negligible compared with adversary training, which involves complex gradient computation.

# F    EXPERIMENTAL DETAILS

In this section, we provide a concrete description of our experiments and detailed hyper-parameters of BATTLE. For each run of experiments, we run on a single Nvidia Tesla V100 GPUs and 16 CPU cores (Intel Xeon Gold 6230 CPU @ 2.10GHz) for training.

## F.1    TASKS

In phase one of our experiments, we evaluate our method on eight robotic manipulation tasks obtained from Meta-world (Yu et al., 2020). These tasks serve as a representative set for testing the effectiveness of our approach. In phase two, we further assess our method on two locomotion tasks sourced from Mujoco (Todorov et al., 2012). By including tasks from both domains, we aim to demonstrate the versatility and generalizability of our approach across different task types. The specific tasks we utilize in our experiments are as follows:

**Meta-world**

- Door Lock: An agent controls a simulated Sawyer arm to lock the door.

- Door Unlock: An agent controls a simulated Sawyer arm to unlock the door.

- Drawer Open: An agent controls a simulated Sawyer arm to open the drawer to a target position.

- Drawer Close: An agent controls a simulated Sawyer arm to close the drawer to a target position.

- Faucet Open: An agent controls a simulated Sawyer arm to open the faucet to a target position.

- Faucet Close: An agent controls a simulated Sawyer arm to close the faucet to a target position.

- Window Open: An agent controls a simulated Sawyer arm to open the window to a target position.

- Window Close: An agent controls a simulated Sawyer arm to close the window to a target position.

**Mujoco**

- Half Cheetah: A 2-dimensional robot with nine links and eight joints aims to learn to run forward (right) as fast as possible.

- Walker: A 2-dimensional two-legged robot aims to move in the forward (right).

### F.2 Hyper-parameters Setting

We adopt the PEBBLE algorithm as our baseline approach for SA-RL (Zhang et al., 2021), and we keep the same parameter settings and neural network structure as described in their work. The specific hyperparameters for SA-RL are provided in Table 4. Similarly, for PA-AD (Sun et al., 2022), we use identical hyperparameter values to those of SA-RL, ensuring a fair comparison between the two methods.

Table 3: Hyper-parameters of BATTLE for adversary training.

| Hyper-parameter | Value | Hyper-parameter | Value |
|---|---|---|---|
| Number of layers | 3 | Hidden units of each layer | 256 |
| Learning rate | 0.0003 | Batch size | 1024 |
| Length of segment | 50 | Number of reward functions | 3 |
| Frequency of feedback | 5000 | Feedback batch size | 128 |
| Adversarial budget | 0.1 | $(\beta_1, \beta_2)$ | $(0.9, 0.999)$ |

Table 4: Hyper-parameters of SA-RL for adversary training.

| Hyper-parameter | Value | Hyper-parameter | Value |
|---|---|---|---|
| Number of layers | 3 | Hidden units of each layer | 256 |
| Learning rate | 0.00005 | Mini-Batch size | 32 |
| Length of segment | 50 | Number of reward functions | 3 |
| Frequency of feedback | 5000 | Feedback batch size | 128 |
| Adversarial budget | 0.1 | Entropy coefficient | 0.0 |
| Clipping parameter | 0.2 | Discount $\gamma$ | 0.99 |
| GAE lambda | 0.95 | KL divergence target | 0.01 |

### F.3 Victim Setting

Our experiment is divided into two phases. In the first phase, we conduct experiments using a variety of simulated robotic manipulation tasks from the Meta-world environment. In the second phase, we shift our focus to two continuous control environments from the OpenAI Gym MuJoCo suite.

**Meta-world.** We train the victim models on the Meta-world tasks using the SAC (Soft Actor-Critic) algorithm proposed by Haarnoja et al. (2018). We employ a fully connected neural network as the

policy network for the SAC algorithm. The detailed hyperparameters used in our experiments are provided in Table 5.

Table 5: Hyper-parameters of SAC for victim training.

| Hyper-parameter | Value | Hyper-parameter | Value |
|---|---|---|---|
| Number of layers | 3 | Initial temperature | 0.1 |
| Hidden units of each layer | 256 | Optimizer | Adam |
| Learning rate | 0.0001 | Critic target update freq | 2 |
| Discount $\gamma$ | 0.99 | Critic EMA $\tau$ | 0.005 |
| Batch size | 1024 | $(\beta_1, \beta_2)$ | $(0.9, 0.999)$ |
| Steps of unsupervised pre-training | 9000 | Discount $\gamma$ | 0.99 |

**Mujoco.** We directly utilize the well-trained model for demonstrating the vulnerability of the Decision Transformer. Specifically, we use the Cheetah agent[4] and the Walker agent[5] with expert-level.

### F.4 SCENARIO DESIGNING

To validate the effectiveness of our approach, we carefully designed two experimental scenarios: the Manipulation Scenario and the Opposite Behavior Scenario. In the Manipulation Scenario, the victim policy is a well-trained policy on robotic tasks. The objective of the adversary is to manipulate the agent's behavior through targeted adversarial attacks, causing the agent to grasp objects that are far from the original target location. The successful execution of such grasping actions indicates the success of the adversarial attack. In the Opposite Behavior Scenario, the victim policy is a well-trained policy on simulated robotic manipulation tasks. The goal of the attacker is to redirect the agent's behavior towards tasks that are opposite in nature to the original objective. For instance, if the victim policy is designed to open windows, the attacker aims to modify the agent's behavior to close the windows instead.

Table 6: Success rate of different methods with varying numbers of preference labels on the Drawer Open task in the manipulation scenario and the Faucet Close task in the opposite behavior scenario. The success rate is reported as the mean and standard deviation over 30 episodes.

| Environment | Feedback | BATTLE (ours) | PA-AD | SA-RL |
|---|---|---|---|---|
| **Drawer Open** (manipulation) | 3000 | $65.7\% \pm 37.1\%$ | $0.0\% \pm 0.0\%$ | $8.3\% \pm 13.2\%$ |
| | 5000 | $86.7\% \pm 18.1\%$ | $0.0\% \pm 0.0\%$ | $21.3\% \pm 18.9\%$ |
| | 7000 | $95.7\% \pm 13.6\%$ | $0.0\% \pm 0.0\%$ | $28.0\% \pm 28.1\%$ |
| | 9000 | $97.0\% \pm 6.9\%$ | $0.0\% \pm 0.0\%$ | $13.0\% \pm 18.5\%$ |
| **Faucet Close** (opposite behavior) | 1000 | $69.7\% \pm 35.2\%$ | $16.7\% \pm 9.4\%$ | $2.0\% \pm 6.0\%$ |
| | 3000 | $79.0\% \pm 16.2\%$ | $29.0\% \pm 14.0\%$ | $6.0\% \pm 11.7\%$ |
| | 5000 | $95.3\% \pm 9.2\%$ | $21.3\% \pm 12.8\%$ | $3.3\% \pm 12.7\%$ |
| | 7000 | $95.3\% \pm 7.6\%$ | $22.7\% \pm 12.4\%$ | $4.0\% \pm 7.1\%$ |

## G EXTENSIVE EXPERIMENTS

**Impact of Feedback Amount.** We evaluate the performance of BATTLE using different numbers of preference labels. Table 6 presents the results of all methods with varying numbers of labels: $3000, 5000, 7000, 9000$ for the Drawer Open task in the manipulation scenario and $1000, 3000, 5000, 7000$ for the Faucet Close task in the opposite behavior scenario. Based on the experimental results shown in Table 6, we conclude that providing an adequate amount of human feedback improves the performance of our method, leading to a stronger adversary and a more stable attack success rate. We observe that the performance of BATTLE consistently improves as the number of preference labels increases, highlighting the crucial impact of the number of preference labels on

---

[4]https://huggingface.co/edbeeching/decision-transformer-gym-halfcheetah-expert
[5]https://huggingface.co/edbeeching/decision-transformer-gym-walker2d-expert

adversary learning. In contrast, SA-RL and PA-AD exhibit poor performance even with a sufficient amount of human feedback, with PA-AD failing entirely in the manipulation scenario. This can be attributed to the limited exploration space of these methods, which is constrained by the fixed victim policy. In contrast, BATTLE achieves better exploration by incorporating an intention policy, resulting in improved performance.

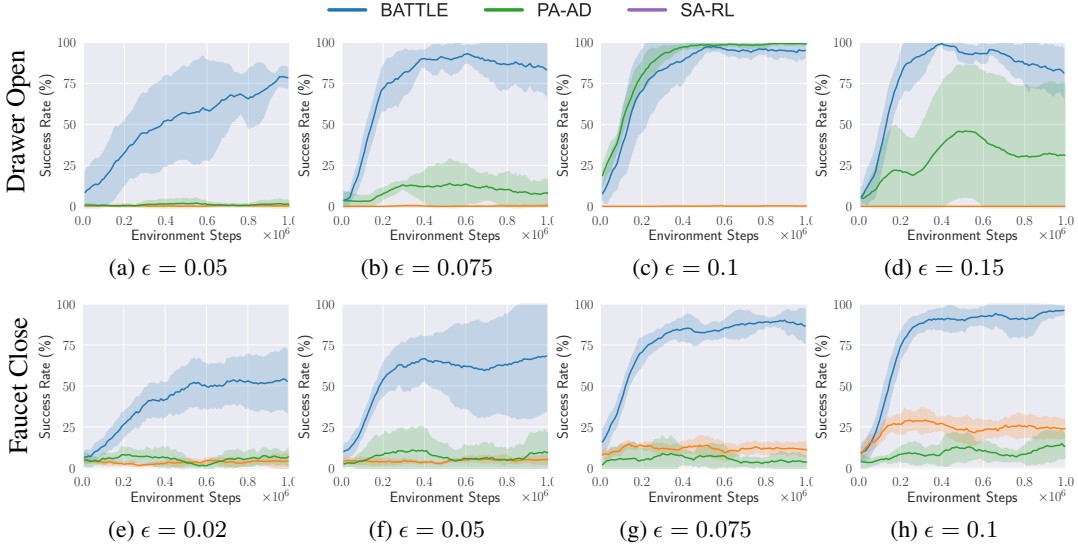

Figure 8: Training curves of success rate with different adversarial budgets on Drawer Open for the manipulation scenario and Faucet Close for the opposite behavior scenario. The solid line and shaded area denote the mean and the standard deviation of the success rate across five runs.

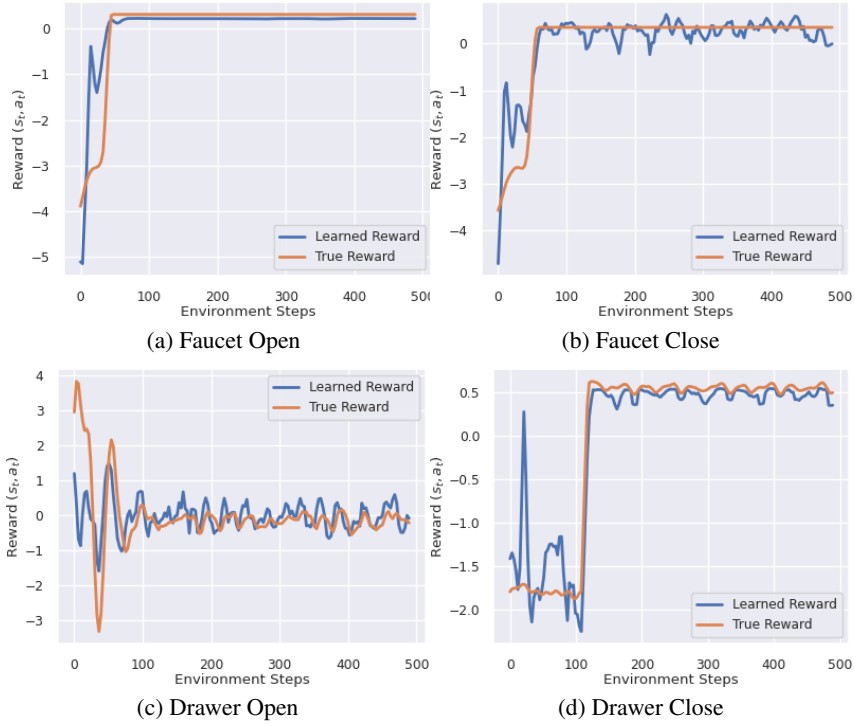

Figure 9: Time series of the normalized learned reward (blue) and the ground truth reward (orange). These rewards are obtained from rollouts generated by a policy optimized using BATTLE.

**Impact of Different Attack Budgets.** We also investigate the impact of the attack budget on the performance. To gain further insights, we conduct additional experiments with different attack budgets: $0.05, 0.075, 0.1, 0.15$ for the Drawer Open task and $0.02, 0.05, 0.075, 0.1$ for the Faucet Close task in the respective scenarios. In Figure 8, we present the performance of the baseline method and BATTLE with different attack budgets. The experimental results demonstrate that the performance of all methods improves with an increase in the attack budget.

**Quality of learned reward functions.** We further analyze the quality of the reward functions learned by BATTLE compared to the true reward function. In Figure 9, we present four time series plots that depict the normalized learned reward (blue) and the ground truth reward (orange). These plots represent two scenarios: opposite behaviors and manipulation tasks. The results indicate that the learned reward function aligns well with the true reward function derived from human feedback. This alignment is achieved by capturing various human intentions through the preference data.

**Robust Agents Training and Evaluating.** An intuitive application of BATTLE is in evaluating the robustness of a given model or enhancing the robustness of an agent through adversarial training. ATLA (Zhang et al., 2021) is a general training framework for improving robustness, which involves alternating training between an agent and an adversary. Building upon this concept, we introduce BATTLE-ATLA, which combines BATTLE with the ATLA framework by training an agent and a BATTLE attacker alternately. The robustness performance of BATTLE-ATLA for a SAC agent is presented in Table 7 and compared with state-of-the-art robust training methods. The experimental results provide two key insights: firstly, BATTLE-ATLA significantly enhances the robustness of agents, demonstrating its effectiveness in improving agent resilience to adversarial attacks. Secondly, BATTLE exhibits the capability to launch stronger attacks on robust agents, highlighting its effectiveness as an adversary in the adversarial training process.

Table 7: Average episode rewards $\pm$ standard deviation of robust agents under different attack methods, and results are averaged across 100 episodes.

| Task | Model | BATTLE | PA-AD | SA-RL | Average Reward |
|---|---|---|---|---|---|
| Door Lock | BATTLE-ATLA | 874±444 | 628±486 | 503±120 | **668** |
| | PAAD-ATLA | 491±133 | 483±15 | 517±129 | 497 |
| | SARL-ATLA | 469±11 | 629±455 | 583±173 | 545 |
| Door Unlock | BATTLE-ATLA | 477±203 | 745±75 | 623±60 | **615** |
| | PAAD-ATLA | 398±12 | 381±11 | 398±79 | 389 |
| | SARL-ATLA | 393±36 | 377±8 | 385±26 | 385 |
| Faucet Open | BATTLE-ATLA | 442±167 | 451±96 | 504±55 | 465 |
| | PAAD-ATLA | 438±53 | 588±222 | 373±32 | 466 |
| | SARL-ATLA | 610±293 | 523±137 | 495±305 | **522** |
| Faucet Close | BATTLE-ATLA | 1048±343 | 1223±348 | 570±453 | **947** |
| | PAAD-ATLA | 661±279 | 371±65 | 704±239 | 538 |
| | SARL-ATLA | 1362±149 | 688±196 | 426±120 | 825 |

