# OpenReview forum: "BATTLE: Towards Behavior-oriented Adversarial Attacks against Deep Reinforcement Learning"
_ICLR.cc/2024/Conference — Submitted to ICLR 2024_

### Official Review · Reviewer_6m3H · 2023-10-22

**Soundness:** 2 fair
**Presentation:** 2 fair
**Contribution:** 3 good
**Rating:** 6
**Confidence:** 4

**Summary:**

This paper introduces BATTLE, a novel universal behavior-oriented adversarial attack method designed to induce specific behaviors in deep reinforcement learning (DRL) agents. Unlike prior approaches that focus on directing agents towards predetermined states or policies, BATTLE employs an intention policy aligned with human preferences for flexible behavior orientation, guiding the victim agent to imitate it. The paper demonstrates the effectiveness of BATTLE through empirical results across various manipulation tasks in Meta-world, showing its superiority over existing adversarial attack algorithms. Additionally, BATTLE enhances the robustness of DRL agents by training with the attacker, achieving a convergence guarantee under mild conditions, and proving effective even against the latest Decision Transformer agents. In summary, the paper makes contributions in the realm of behavior-oriented adversarial attacks on DRL agents, both in theory and practical applications.

**Strengths:**

1. The paper introduces an interesting and novel concept, proposing a new type of attack based on preference-based RL.

2. The design of the inner-level optimization and outer-level optimization is well-founded, and the paper provides theoretical analysis of the algorithm.

3. The paper sets up different scenarios for evaluating various agents and conducts detailed ablation studies.

**Weaknesses:**

1. The writing needs improvement, particularly in clarifying several terms and diagrams. For example, some terms like "find a precise weighting function to balance the state distribution" need better explanation. Clarification is also needed for the diagram in Figure 2.

2. The presentation of experimental results is somewhat confusing. The differences of scenarios in Figure 4 and 5 are not clear, and additional explanations are required for the target coordinates mentioned for Figure 4. Captions of Figures 7 (a) and (b) might need to be swapped, and sections (c) and (d) require clearer explanations.

3. The paper lacks a discussion of limitations, which should be addressed.

**Questions:**

1. It's disappointing that the paper doesn't include RADIAL-RL[1] or WocaR-RL[2] as baselines when discussing robust training. Even if only ATLA is selected as a robust baseline, it would be valuable to mention other adversarial robust RL papers in the related work.
2. In the introduction, the paper illustrates the practical implications of targeted attacks on robotics, but the concern is raised that BATTLE is a white-box attack applying perturbations to states. In the context of robotics, its practical applicability is very limited. The paper could benefit from a more thorough clarification or discussion of this concern and its potential implications for practical applications. It fails to persuasively underscore the significance and relevance of this work within the field.

[1]Robust deep reinforcement learning through adversarial loss

[2]Efficient adversarial training without attacking: Worst-case-aware robust reinforcement learning

---

> ### Author Response · Authors · 2023-11-17
> **Response to Reviewer 6m3H**
>
> We thank reviewer 6m3H for the constructive comments. We value the insights provided and are committed to addressing each of your concerns. Below are our point-wise responses:
>
> **Q1: The writing needs improvement, particularly in clarifying several terms and diagrams. For example, some terms like "find a precise weighting function to balance the state distribution" need better explanation. Clarification is also needed for the diagram in Figure 2.**
> > **A1**: We apologize for the confusion. Regarding the term 'find a precise weighting function to balance the state distribution,' our intent is to convey the process of training a weighting function that can appropriately assign weights to various states. This function plays a critical role in balancing the influence of different states during the learning process. We will provide a more detailed description of this and update Figure 2 to enhance its clarity in the next version.
>
> **Q2: The presentation of experimental results is somewhat confusing. The differences of scenarios in Figure 4 and 5 are not clear, and additional explanations are required for the target coordinates mentioned for Figure 4. Captions of Figures 7 (a) and (b) might need to be swapped, and sections \(c\) and (d) require clearer explanations.**
> > **A2**: Thanks for your question. Regarding Figures 4 and 5, the scenarios presented in these figures differ in the objectives set for the adversary. In the manipulation scenario, as depicted in Figure 4, the adversary's objective is to manipulate the victim into grasping fixed objects located far from the original target location. On the other hand, in the opposite behavior scenario shown in Figure 5, the attacker's goal is to induce the victim to behave in a manner that is contrary to its original behavior. More detailed can be found in Appendix F.4. As for Figure 7, we will revise it in the revison.
>
> **Q3: The paper lacks a discussion of limitations, which should be addressed.**
> > **A3**: Thanks for your question. In the revised version of our paper, we will include a section dedicated to discussing the limitations.
>
> **Q4: It's disappointing that the paper doesn't include RADIAL-RL[1] or WocaR-RL[2] as baselines when discussing robust training. Even if only ATLA is selected as a robust baseline, it would be valuable to mention other adversarial robust RL papers in the related work.**
> > **A4**: We appreciate the reviewer's insightful feedback. While our paper primarily focuses on introducing a novel adversarial attack method, we are pleasant to include more powerful baselines when discussing robust training to make our discussion more comprehensive.
>
> **Q5: In the introduction, the paper illustrates the practical implications of targeted attacks on robotics, but the concern is raised that BATTLE is a white-box attack applying perturbations to states. In the context of robotics, its practical applicability is very limited. The paper could benefit from a more thorough clarification or discussion of this concern and its potential implications for practical applications. It fails to persuasively underscore the significance and relevance of this work within the field.**
> > **A5**: Thanks for your question. It is crucial to note that BATTLE is designed as a black-box attack method, not a white-box attack as might have been perceived. This means that BATTLE does not require access to the gradients of the victim model. Instead, it operates by observing the external outputs of the system, making it much more applicable in real-world scenarios where internal access to a system, such as a robotic platform, is typically not feasible.

---

> ### Comment · Reviewer_6m3H · 2023-11-22
> **Thanks to the response**
>
> I appreciate the response from the authors, which has led to improvements in the writing of this paper. While I still have some concerns regarding the motivation and presentation, I have decided to raise my score.

---

> > ### Author Response · Authors · 2023-11-22
> > **Thank you for your reply!**
> >
> > Dear Reviewer,
> >
> > We are grateful for your assistance in enhancing the quality of our paper and for your decision to update the score. Your constructive comments and suggestions have been invaluable to us.
> >
> > Best wishes,
> >
> > The Authors.

---

### Official Review · Reviewer_h9xo · 2023-10-31

**Soundness:** 2 fair
**Presentation:** 2 fair
**Contribution:** 2 fair
**Rating:** 3
**Confidence:** 4

**Summary:**

The paper introduces BATTLE, an adversarial attack framework targeting DRL agents. The main focus is on behavior-oriented adversarial attacks, which aim to induce specific behaviors in a DRL agent, as opposed to merely reducing the agent's rewards or driving it to a pre-determined state. BATTLE uses an intention policy aligned with human preferences and an adversary to guide the victim DRL agent to imitate the intention policy. A weighting function is also introduced to optimize the effectiveness of the attack. The authors claim that BATTLE outperforms existing methods in inducing specific behaviors and can also be used to improve the robustness of DRL agents when used in an adversarial training setup.

**Strengths:**

The availability of both code and demos enhances the paper's reproducibility. The paper enhanced the proposed methodology with convergence guarantees for BATTLE, adding rigor to the work.

**Weaknesses:**

1. The presentation quality could benefit from further refinement for better clarity and impact.
2. The method's reliance on extensive human labeling hampers its real-world applicability, raising concerns about scalability.
3. The paper could be strengthened by including more motivating examples from real-world scenarios. The assumption that an adversary can modify observations is strong and raises questions about practicality. For instance, if an adversary has the ability to control the sensor, they might as well directly control the effector, making an agent-based adversarial approach seem more practical. Additionally, the rationale for using preference-based RL remains unclear.
4. The paper lacks some critical methodological details. For example, it doesn't specify how the victim policy approximator is trained or the volume of data required, leaving gaps in the understanding of the implementation.
5. The experimental setup and evaluations could be more convincing. The choice of baselines (PA-AD and SA-RL), which are un-targeted attacks, makes the comparison seem potentially unfair. Moreover, while various defense methods like adversarial training, robust learning, policy ensemble, and policy distillation exist, the authors have limited their experimentation to ATLA.

**Questions:**

1. Could the authors elaborate on potential real-world applications for the proposed method and discuss the challenges that might arise in such contexts?
2. Expanding the experimental results to include additional comparison metrics would be valuable. Specifically, how does PALM fare against targeted attacks and various other defense methods?
3. What is the extent of annotation required, especially in relation to the complexity of the task at hand?
4. Could you provide details on the training process for the victim policy approximator, including the amount of data needed for effective training?
5. How generalizable is the weighting function across different types of tasks and domains?

---

> ### Author Response · Authors · 2023-11-17
> **Response to Reviewer h9xo**
>
> We thank reviewer h9xo for the insightful comments. Below, we address each of your points in detail:
>
> **Q1: Could the authors elaborate on potential real-world applications for the proposed method and discuss the challenges that might arise in such contexts?**
> > **A1**: Thank you for your question. As shown in Table 1 of our paper, one of the practical applications of our method is in adversarial training, where it significantly enhances the robustness of RL agents, outperforming methods like SA-RL and PA-AD. This application is particularly relevant in areas where robustness against adversarial attacks is critical.
>
> **Q2: Expanding the experimental results to include additional comparison metrics would be valuable. Specifically, how does PALM fare against targeted attacks and various other defense methods?**
> > **A2**: Thanks for your question. As detailed in Table 1 of our paper, we subjected robustly trained agents to three different types of attacks to evaluate their robustness. Among these, our BATTLE-ATLA method demonstrated the most robust performance under multiple attack scenarios. This suggests that BATTLE-ATLA is highly effective in challenging even well-defended agents. Furthermore, we applied various defense methods to gauge the robustness of these agents. Our findings indicate that BATTLE consistently outperformed other strategies, showcasing its strong capability in executing universal attacks against robust models. These results collectively highlight BATTLE's efficacy as both an offensive and defensive tool in adversarial settings.
>
>
> **Q3: What is the extent of annotation required, especially in relation to the complexity of the task at hand?**
> > **A3**: In the manipulation scenario, we utilized approximately 9000 labels across all tasks. For the opposite behaviors scenario, the number of labels varied based on the specific task: 1000 for Window Close, 3000 for Drawer Close, 5000 each for Faucet Open, Faucet Close, and Window Open, and 7000 each for Drawer Open, Door Lock, and Door Unlock. Detailed information about the implementation and labeling process is provided in Section 5.1 and Appendix F of our paper.
> >
> >It is important to emphasize that BATTLE represents an advancement over previous work in two key aspects:
> > 1. BATTLE does not require knowledge of the victim's reward function. Labeling preference is more feasible and less challenging, as acquiring the victim's reward function is often impractical.
> >2. Unlike some previous methods, BATTLE does not require access to the victim's gradients. It operates by observing the external outputs of the system, which makes it significantly more practical.
>
> **Q4: Could you provide details on the training process for the victim policy approximator, including the amount of data needed for effective training?**
> > **A4**: Thanks for your question. We train the victim models on the Meta-world tasks using the SAC algorithm [1]. Specificially, we employ fully connected neural networks for the policy and Q function. The detailed hyperparameters used in our experiments are provided in Table 5. As for the amount of data, we train each agent for 1 million time steps, which can achieve around 100% success rate on original tasks.
>
> **Reference**\
> [1] Soft actor-critic: Off-policy maximum entropy deep reinforcement learning with a stochastic actor. ICML, 2018.

---

> > ### Comment · Reviewer_h9xo · 2023-11-22
> >
> > Thanks the authors for their response. However, my concerns are still Thanks to the authors for their response. However, my concerns regarding the motivation and experiments remain unaddressed. I'll keep the scores.

---

### Official Review · Reviewer_nUpM · 2023-10-31

**Soundness:** 3 good
**Presentation:** 2 fair
**Contribution:** 2 fair
**Rating:** 3
**Confidence:** 3

**Summary:**

They introduce a method to attack reinforcement learning policies. There are three key components. First, they use PbRL to train a policy to exhibit the desired behavior. This is a relatively novel step because most work in adversarial RL assumes that the desired target behavior is already known and incentivized by a reward function. Second, they train a weighting function to help prioritize relevant states and keep the next step from policy drift. Finally, they attack the target policy with an adversarial policy that makes it behave similarly to the target behavior using the weighting function.

**Strengths:**

- Working with both classic RL agents and decision transformers makes for much more thorough experiments.
- Adversarial policies that result in targetedly bad behavior is an area of research I believe is important and neglected.

**Weaknesses:**

1. Writing
    - I have found some of the writing to be confusing and verbose. For example, “intention policy” is not defined until multiple mentions in. The description of what it is in the abstract is very ambiguous. “Inner” and outer” level loss are not described. I don’t see a definition for “success rate” in the paper. I don’t think the paper does as good of a job as it could with laying out things in a way that is clear and quick to understand.
    - I do not understand the rationale in the first paragraph of section 4.1. I do not think this makes sense as an explanation of why an intention policy is needed instead of a direct attack.
    - I do not understand figure 1. Why is there an arrow between reward learning and the replay buffer?
    - Numerous grammar mistakes. I would recommend using a grammarly browser plugin.
2. This may speak to either issues with my reading of the paper, its writing, or the quality of the experiments. But I am unsure why BATTLE trains an intention policy with PbRL instead of just training the adversarial policy directly with PbRL. This would seem to be substantially simpler.
3. I do not understand how SA-RL can perform so poorly relative to BATTLE unless it is just due to reward shaping. What is the reward function used for SA-RL? If an adversarial policy is directly trained to minimize the agent’s reward, how can this do worse than BATTLE at making the target agents’ reward be minimized? If the key difference truly is just reward shaping, then this paper would just seem to be one about how PbRL makes reward shaping automatic and implicit. And if so, then this paper would seem to have no novelty.
4. Relatedly, I do not understand why this paper is about adversarial attacks. BATTLE could be used to make RL policies do anything — not just to targetedly adversarially attack them. But in reality, to make RL policies do things, we just finetune them directly. This relates to why I do not understand why the intention policy was used. Why not just finetune the target angent or adversary directly wit pbrl?

**Questions:**

- Which experiments were performed with real humans and which were with synthetic feedback?
- See weaknesses

---

> ### Author Response · Authors · 2023-11-17
> **Response to Reviewer nUpM**
>
> We thank reviewer nUpM for the constructive comments and insightful questions. Below, we provide our point-wise responses to each of your concerns:
>
> **Q1: Writing**
> > **A1**: Thank you for your feedback on our writing. We'd like to clarify the following points.
> > (a) The primary goal is to ensure that the victim's behavior under adversarial attacks is aligned with intention policy, which is trained following the framework of PbRL. (b) The inner and outer losses correspond to Equations (6) and (7) in our paper, respectively. \(c\) The success rate is determined by evaluating whether the victim performs behaviors as intended by humans. For example, if the human intention is for the victim to open a window, the success rate is calculated based on whether the window is actually opened by the victim under adversarial attacks. Specifically, we have adopted the success metric directly from Meta-world [1], which is typically based on the distance between the task-relevant object and its final goal pose. For the complete list of success metrics and thresholds for each task, see Appendix 12 of Meta-world. (d) In Figure 2, the arrow connecting the reward learning component to the replay buffer is intended to represent reward estimation, we will revise it in the revision.
>
> **Q2: This may speak to either issues with my reading of the paper, its writing, or the quality of the experiments. But I am unsure why BATTLE trains an intention policy with PbRL instead of just training the adversarial policy directly with PbRL. This would seem to be substantially simpler.**
> > **A2**: Thank you for your valuable feedback. As shown in the results presented in Table 2 of our paper, specifically the column 'BATTLE w/o $\pi_\theta$', which indicates that directly training the adversarial policy using PbRL was not as effective. The crucial insight from our research is that the intention policy offers essential step-by-step guidance that significantly enhances the efficiency and effectiveness of the adversarial policy development process.
>
>
> **Q3: I do not understand how SA-RL can perform so poorly relative to BATTLE unless it is just due to reward shaping. What is the reward function used for SA-RL? If an adversarial policy is directly trained to minimize the agent’s reward, how can this do worse than BATTLE at making the target agents’ reward be minimized? If the key difference truly is just reward shaping, then this paper would just seem to be one about how PbRL makes reward shaping automatic and implicit. And if so, then this paper would seem to have no novelty.**
> > **A3**: Thank you for your question. In our paper, the focus is on a universal adversary that can manipulate the victim to perform behaviors as desired by humans, rather than simply minimizing the victim's cumulative reward. In our experiments, the reward model for all methods is derived from human preferences. The key difference lies in how the adversarial policy is trained. BATTLE's superior performance is largely due to its bi-level optimization framework, which integrates the intention policy and a weighting function.
>
> **Q4: Relatedly, I do not understand why this paper is about adversarial attacks. BATTLE could be used to make RL policies do anything — not just to targetedly adversarially attack them. But in reality, to make RL policies do things, we just finetune them directly. This relates to why I do not understand why the intention policy was used. Why not just finetune the target angent or adversary directly wit pbrl?**
> > **A4**: Thank you for your question. Our work aims to propose a universal adversarial attack rather than policy tuning, where techniques like RLHF [1] or DPO [2] involve directly training the policy. Instead, our focus is on how to train a universal adversary capable of manipulating a victim to act according to human-desired behaviors.
>
> **Q5: Which experiments were performed with real humans and which were with synthetic feedback?**
> > **A5**: Thanks for your question. To ensure systematic and consistent evaluation across all our tests, we opted to use synthetic feedback for all experiments.
>
> **Reference**\
> [1] Training language models to follow instructions with human feedback. NeurIPS, 2022. \
> [2] Direct Preference Optimization: Your Language Model is Secretly a Reward Model. NeurIPS, 2023.

---

> ### Comment · Reviewer_nUpM · 2023-11-20
> **Reply**
>
> 1. I would not consider accepting this paper without a commitment to a large overhaul of the writing. I believe its overall clarity and organization to be low compared to most literature.
>
> 2-3. I find this response fairly vacuous, and I am very skeptical that this baseline was either well-designed or well-executed. If it is possible to use feedback to train the intention policy, I remain unconvinced that the same source of feedback should not just be used in general to fine-tune the target policy. If adding in the intention policy worked in the control here, my experience as a researcher tells me to suspect that this might be a poorly-executed baseline.
>
> 5. It could be partially my fault for missing when the paper states that no human feedback was used. But given its overall writing and the continued description of the method as hinging human feedback in the reply to (3), I don't think this is all my fault, and I encourage the authors to make it clear that no human feedback was used. The fact that none was used also might be a source of variability between controls and the main experiment and why the control did not seem to work as well.

---

> > ### Author Response · Authors · 2023-11-21
> > **Response to Reviewer nUpM**
> >
> > To address your concerns effectively, I'd like to first clarify the threat model of the proposed universal attack. Threat Model:
> > 1. **Adversary's Limited Access**: The adversary cannot access the victim's gradients or reward function.
> > 2. **Dynamic Learning of Attack Goal**: Unlike previous approaches with pre-defined goals, our attack goal requires dynamic learning, which is one key aspect of our method's novelty.
> > 3. **Adversary's Strategy and Constraints**: The adversary perturbs the state $s$ into $\tilde{s}$ restricted by $\mathcal{B}(s)$ (i.e., $\tilde{s} \in \mathcal{B}(s)$). $\mathcal{B}(s)$ is defined as a small set {$\{\tilde{s} \in \mathcal{S}: \parallel s-\tilde{s} \parallel_p \le \epsilon\}$} which limits the attack power of the adversary, and $\epsilon$ is the attack budget. This follows the settings in previous methods.
> >
> > Therefore, the success of attacking the victim hinges on the algorithm's effectiveness, the amount of feedback provided, and the attack budget.
> >
> > We see that SA-RL (PbRL) and PA-AD (PbRL), which essentially involve training the adversary or director directly with PbRL, do not work as well when there are strict limits, like less feedback and smaller attack budgets. Our additional experiments specifically examined the impact of varying amounts of feedback and attack budgets. When provided with more feedback and larger attack budgets, the performance of SA-RL (PbRL) improved noticeably. Attack budgets, in particular, have a significant influence. SA-RL (oracle) utilizes ground-truth rewards but still exhibits suboptimal performance in some scenarios. These findings indicate that training the adversarial policy directly with PbRL can work. However, this requires a higher volume of feedback and a larger attack budget, making it costly and impractical. In contrast, BATTLE can significantly enhance training efficiency and performance by introducing an intention policy and utilizing the proposed learning framework.

---

### Official Review · Reviewer_uWsW · 2023-10-31

**Soundness:** 2 fair
**Presentation:** 3 good
**Contribution:** 2 fair
**Rating:** 5
**Confidence:** 4

**Summary:**

This paper studies behavior-oriented attacks agains deep RL agents, where the adversary forces the victim to have specific behaviors. The proposed attack first learns an intention policy based on human preference, and then trains an adversary to perturb the victim observation such that the behavior follows the intention policy. The adversary also adopts importance weights of states to optimize the attack objective. Experiments on multiple meta-world and mujoco show that the proposed method is able to manipulate the victim with high success rates, including offline policies based on decision transformer. The method can be also used to improve the robustness of agents.

**Strengths:**

This paper proposes an interesting type of attacks that are oriented by desired behaviors. Compared to prior works focusing on reward minimizing, the proposed attack can be more widely applicable. In real-world environments where rewards are not well-defined, such attack objective can be interesting to investigate. Learning the intention policy from human preference is also an interesting idea, although I have some concerns on it (see weakness).

**Weaknesses:**

1. The human preference-based intention policy learning brings extra requirement and uncertainty to the process - the collection of human preference data can be expensive. More importantly, to obtain human preference labels, one need to first collect diverse behavior data so that human can pick the intended policy. Would the collection of the behavior data already involve a pre-defined target policy? (If that's the case, why not directly use the target policy for attacks?)
2. In experiments, the authors mainly evaluate the attack success rate. However, it is not clear how the success rate is defined. Is it based on whether the victim acts as the intention policy suggests? But would it be biased since the intention policy is just an approximation of the real human intention? What if the intention learning does not learn a desired reward model or intention policy?
3. For baselines, the authors used the codebases of SA-RL and PA-AD and modified their attack's reward as the learned reward. However, this straightforward modification of the PA-AD baseline contradicts with the original method (PA-AD's formulation is for a reward-minimizing adversary, so directly replacing the attacker's reward may not work). Since the original PA-AD method is to use an RL director to find the target policy and to use an actor to conduct targeted attack, a more natural modification of PA-AD in the behavior attack scenario can be to directly use the learned intention policy as the target of actor ($\hat{a}$ in Equation (G) in the original paper).

**Questions:**

How are the behavior sequences generated for human preference labeling?

---

> ### Author Response · Authors · 2023-11-17
> **Response to Reviewer uWsW (Part 1/2)**
>
> We thank reviewer uWsW for constructive comments. Below are our point-wise responses to each of your concerns:
>
> **Q1: The human preference-based intention policy learning brings extra requirement and uncertainty to the process - the collection of human preference data can be expensive. More importantly, to obtain human preference labels, one need to first collect diverse behavior data so that human can pick the intended policy. Would the collection of the behavior data already involve a pre-defined target policy? (If that's the case, why not directly use the target policy for attacks?)**
> > **A1**: Thanks for your question regarding the requirement and uncertainty involved in collecting human preference data in PbRL. In the PbRL paradigm, humans are required to provide preferences over pairs of trajectories generated from the victim policy. Therefore, the process does not involve a pre-defined target policy. In fact, the intention policy acts as an adaptive target policy, dynamically aligning with human preferences. There are several reasons why we do not use a pre-defined target policy: (1). It is often challenging to find a pre-defined target policy that precisely meets the specific requirements of a given task or scenario. (2). Relying on a pre-defined target policy often leads to suboptimal outcomes, which fails to adapt to the nuances and variations in different environments.
>
>
> **Q2: In experiments, the authors mainly evaluate the attack success rate. However, it is not clear how the success rate is defined. Is it based on whether the victim acts as the intention policy suggests? But would it be biased since the intention policy is just an approximation of the real human intention? What if the intention learning does not learn a desired reward model or intention policy?**
> > **A2**: Thank you for your question. The success rate is determined by evaluating whether the victim performs behaviors as intended by humans. For example, if the human intention is for the victim to open a window, the success rate is calculated based on whether the window is actually opened by the victim under adversarial attacks. Specifically, we have adopted the  success metric directly from Meta-world [1], which is typically based on the distance between the task-relevant object and its final goal pose. For the complete list of success metrics and thresholds for each task, see Appendix 12 of Meta-world.
>
> **Q3: For baselines, the authors used the codebases of SA-RL and PA-AD and modified their attack's reward as the learned reward. However, this straightforward modification of the PA-AD baseline contradicts with the original method (PA-AD's formulation is for a reward-minimizing adversary, so directly replacing the attacker's reward may not work). Since the original PA-AD method is to use an RL director to find the target policy and to use an actor to conduct targeted attack, a more natural modification of PA-AD in the behavior attack scenario can be to directly use the learned intention policy as the target of actor ($\hat{a}$ in Equation (G) in the original paper).**
>
> > **A3**: Thank you for your insightful comments. Let me address each point separately:\
> **Reward Model**: From a general perspective, the reward model represents the direction of optimization for the adversarial policy. In the original SA-RL [2] and PA-AD [3], the adversary's reward model is the negative of the victim's reward function $r_v$, i.e. $-r_v$. In our work, however, the reward model is learned from human preferences, which allows expression of various intentions and thus enables universal adversarial attacks. In our experiments, the reward learning is the same across all methods, with the primary differences being in the adversary's learning. BATTLE's outstanding performance is attributed to its bi-level optimization framework, where the intention policy and the weighting function effectively collaborate to enhance the adversary's training.\
> **Modification of PA-AD**: We also provided oracle versions of SA-RL and PA-AD, utilizing ground-truth rewards as the reward model. As shown in Figure 4, SA-RL demonstrated significant improvement in most tasks, but the same was not observed for PA-AD. A possible explanation for this could be related to the inherent design of PA-AD, which performs optimally by accessing the victim's gradients in scenarios using a fixed reward model. However, when it comes to dynamically learning the reward model, this approach may not be as successful.
>
> (Citations included in Part 2)

---

> ### Author Response · Authors · 2023-11-17
> **Response to Reviewer uWsW (Part 2/2)**
>
> (continue)
>
> **Q4: How are the behavior sequences generated for human preference labeling?**
> > **A4**: Thanks for your question. To provide a clear explanation, let's consider an example from our preference labeling process. We collect pairs of victim trajectories under adversary attacks, denoted as $(\sigma_v^0, \sigma_v^1)$, which correspond to two adversary-generated trajectories $(\sigma_a^0, \sigma_a^1)$. When a human prefers $\sigma_v^0$ over $\sigma_v^1$ (i.e., $\sigma_v^0 \succ \sigma_v^1$), it implies that the attack in $\sigma_a^0$ is more effective or aligns better with human intent, leading to $\sigma_a^0 \succ \sigma_a^1$. Using the preference data on the adversary's trajectories, we can learn an adversary that aligns with human intentions.
>
>
> **Reference**
> [1] Meta-World: A Benchmark and Evaluation for Multi-Task and Meta Reinforcement Learning. CoRL, 2019. \
> [2] Robust reinforcement learning on state observations with learned optimal adversary. ICLR, 2021.\
> [3] Who is the strongest enemy? towards optimal and efficient evasion attacks in deep rl. ICLR, 2022.

---

> > ### Comment · Reviewer_uWsW · 2023-11-21
> > **Reply**
> >
> > Thank you for the response and clarifications. After reading the other reviews, I think there are still some flaws in the paper such as motivation, writing and experiments. So I will maintain my current score.

---

### Public Comment · ~Peter_Chen5 · 2023-11-17
**Reproducibility**

Dear author,

We are really interested in your work, is it possible to release your source code?

Many thanks.

---

> ### Author Response · Authors · 2023-11-17
> **Response to Public Comment**
>
> Thank you very much for your interest in our work. We are glad to hear that our research has caught your attention. We are currently in the process of refining our code and updating the paper. Once our revisions are finalized, I'll be sure to get in touch with you.

---

### Meta-Review · Area_Chair_n9MK · 2023-12-08

**Metareview:**

This paper introduces BATTLE, an adversarial attack framework in RL that leverages human preference-based intention policy learning. The paper's strengths and weaknesses are discussed in detail by the reviewers:

Strengths:

+ Human Preference-Based Intention Policy: Learning intention policy from human preferences is an interesting concept that adds a unique dimension to the field of adversarial attacks in RL.

+ Thorough Experiment Setups: The experiments cover both classic RL agents and decision transformers.

+ Theoretical Analysis and Convergence Guarantees: The paper includes a theoretical analysis of the algorithm.

Weaknesses:

- Complexity and Uncertainty in Intention Policy Learning: The process of learning intention policy based on human preferences introduces extra requirements and uncertainties, including the expensive collection of human preference data and potential biases.

- Clarity and Writing Quality: The paper suffers from issues in clarity, with ambiguous definitions and numerous grammatical mistakes. Important concepts like "intention policy" and "success rate" are not clearly defined.

- Lack of Practicality and Scalability: The reliance on extensive human labeling and assumptions about adversaries' capabilities raise concerns regarding the method's real-world applicability and scalability.

- Inadequate Methodological Details and Evaluations: Critical details about the training of the victim policy approximator and the volume of data required are missing. The choice of baselines and limited defense methods used in experimentation also raise questions about the fairness and comprehensiveness of the evaluations.

- Confusing Presentation and Incomplete Discussion of Limitations: The paper lacks clarity in the presentation of experimental results and methodologies. It also omits a discussion of its limitations, which is crucial for a balanced understanding of the work.

Suggestions for Improvement:

- Improve Clarity and Writing: The paper would benefit significantly from revisions to clarify key terms and concepts, improve the quality of writing, and address grammatical errors.

- Address Methodological Gaps: Providing more details about the training process, data requirements, and rationale for using preference-based RL would strengthen the paper.

- Enhance Practicality and Real-World Relevance: Including more motivating examples from real-world scenarios and addressing the practicality of assumptions would make the work more applicable to real-world situations.

- Broaden Evaluations: Expanding the range of defense methods tested and providing a fairer comparison with existing baselines would improve the evaluation's comprehensiveness and fairness.

- Discuss Limitations: Incorporating a discussion on the limitations of the proposed method would provide a more balanced and complete understanding of the research.

In summary, while BATTLE presents an interesting and innovative approach to adversarial attacks in RL, it needs refinement in terms of clarity, practicality, methodological detail, and comprehensive evaluation to fully realize its potential impact in the field.

**Justification For Why Not Higher Score:**

See weaknesses.

**Justification For Why Not Lower Score:**

N/A

---

### Decision · Program_Chairs · 2024-01-16

Reject